# Planning-Driven Programming: A Large Language Model Programming Workflow

## Abstract

The strong performance of large language models (LLMs) on natural language processing tasks raises extensive discussion on their application to code generation. Recent work suggests multiple sampling approaches to improve initial code generation accuracy or program repair approaches to refine the code. However, these methods suffer from LLMs' inefficiencies and limited reasoning capacity. In this work, we propose an LLM programming workflow (LPW) designed to improve both initial code generation and subsequent refinements within a structured two-phase workflow. Specifically, in the solution generation phase, the LLM first outlines a solution plan that decomposes the problem into manageable sub-problems and then verifies the generated solution plan through visible test cases. Subsequently, in the code implementation phase, the LLM initially drafts a code according to the solution plan and its verification. If the generated code fails the visible tests, the plan verification serves as the intended natural language solution to consistently inform the refinement process for correcting bugs. We further introduce SLPW, a sampling variant of LPW, which initially generates multiple solution plans and plan verifications, produces a program for each plan and its verification, and refines each program as necessary until one successfully passes the visible tests. Compared to the state-of-the-art methods across various existing LLMs, our experimental results show that LPW significantly improves the Pass@1 accuracy by up to 16.4% on well-established text-to-code generation benchmarks, especially with a notable improvement of around 10% on challenging benchmarks. Additionally, SLPW demonstrates up to a 5.6% improvement over LPW and sets new state-of-the-art Pass@1 accuracy on various benchmarks, e.g., 98.2% on HumanEval, 84.8% on MBPP, 64.0% on APPS, and 35.3% on CodeContest, using the advanced LLM GPT-4o as the backbone.

## 1 Introduction

Code generation, also known as *program synthesis*, studies the automatic construction of a program that satisfies a specified high-level input requirement (Gulwani et al., 2017). Recently, large language models (LLMs) pre-trained on extensive code-related datasets (Brown et al., 2020; Meta, 2024; Li et al., 2023; Roziere et al., 2023; Achiam et al., 2023; Muennighoff et al., 2023) have shown success in code-related tasks, such as code generation from natural language descriptions, also named as text-to-code generation (Chen et al., 2021; Austin et al., 2021; Li et al., 2022), code translation (Pan et al., 2024; Yang et al., 2024), and code completion (Izadi et al., 2024). However, LLM-based code generation remains challenging due to stringent lexical, grammatical, and semantic constraints (Scholak et al., 2021). To overcome these challenges, multiple initial programs are generated (Chen et al., 2021; Chowdhery et al., 2023), followed by different best-program selection strategies to improve code generation performance over LLMs (Li et al., 2022; Chen et al., 2023a; Zhang et al., 2023; Ni et al., 2023).

Code generation substantially benefits from the empirical insights of human programmers. In practice, human programmers develop high-quality code by consistently identifying and rectifying errors through the analysis of test case executions, rather than a single effort (Huang et al., 2023c; Chen et al., 2023b). Different studies have refined programs based on execution results and LLM-generated information such as code and error explanation (Tang et al., 2023; Shinn et al., 2023; Madaan et al., 2023). Recent work further optimizes refinement (debugging) methods by performing

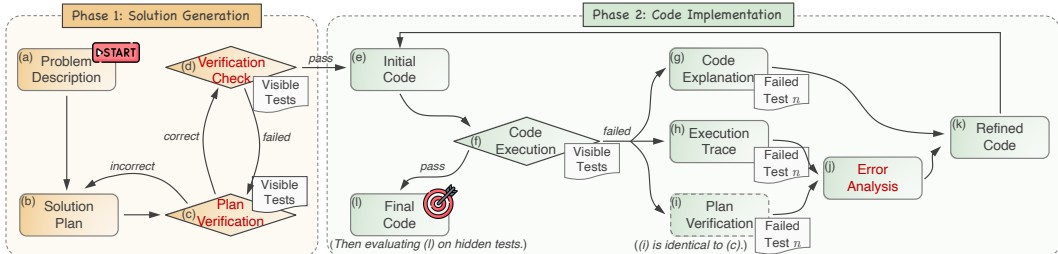

Figure 1: The pipeline of LPW, a *large language model programming workflow*, where the components highlighted in red are exclusive to LPW. LPW consists of two phases. In the solution generation phase, LPW initially creates a solution plan (block (b)) for a problem (block (a)), along with the plan verification (block (c)) for each visible test. If the plan verification infers the accurate output for each visible test based on the solution plan (block (c)) and no incorrect logic is found in the verification check process (block (d)), LPW uses the generated plan and plan verification to help LLMs draft the initial program (block (e)) at the beginning of the code implementation phase. If the initial program passes all visible tests after execution (block (f)), it is used as the final code (block (l)) and then assessed with hidden tests. Otherwise, the LLM-generated code explanation (block (g)) and error analysis (block (j)) serve as debugging inputs to refine the error program (block (k)). The LLM-generated error analysis involves comparing the execution trace (block (h)) with the plan verification (block (i)) on the failed visible test to identify logic flaws in the code implementation and provide repair suggestions. The refined program is reevaluated on the visible tests to determine the necessity for further debugging iterations.

*rubber duck debugging* processes (Chen et al., 2023b) and leveraging control flow graph information to assist LLMs in locating bugs (Zhong et al., 2024).

Software development models such as *Waterfall* and *Scrum* underscore the importance of communication among various development roles in the production of high-quality software (Davis, 2012; Schwaber, 2004; Andrei et al., 2019). Motivated by this principle, several studies (Lin et al., 2024; Qian et al., 2024; Dong et al., 2023b) have employed LLM instances as customized agents, assigning them diverse development roles and facilitating their collaboration. Multi-agent collaborative code generation emphasizes the distinct workload for each LLM agent, e.g., requirement analyst, architect, programmer, and tester (Lin et al., 2024). Additionally, various communication strategies have been proposed to ensure program quality. For example, Hong et al. (2024) introduced a communication protocol to ensure efficient interactions. Qian et al. (2024) described a communicative dehallucination mechanism to encourage high-quality communication among LLM agents.

However, all the aforementioned methods have certain weaknesses. Multiple sampling approaches suffer from sampling inefficiency and conflict with human programming strategies. In code refinement, feedback messages often lack precise correction instructions, leading to numerous refinements that deviate from the intended solution. Additionally, refining programs that significantly diverge from the problem description remains an open challenge (Tian & Chen, 2023). In multi-agent collaborative code generation, ineffective feedback mechanisms degrade communication quality. This issue is exacerbated when an excessive number of agents are involved, resulting in increased token consumption (Huang et al., 2023a).

In this work, we propose LPW, a *large language model programming workflow*, specifically for text-to-code generation, addressing the aforementioned limitations. LPW involves two phases for code generation: the solution generation phase for plan and plan verification creation, and the code implementation phase for initial code development and subsequent refinements. The pipeline of LPW is depicted in Figure 1. LPW leverages various information, including LLM-generated solution plan (Jiang et al., 2023) (block (b)), LLM-generated code explanation (Chen et al., 2023b) (block (g)), and runtime information from program execution (Zhong et al., 2024) (block (h)) to boost the code generation performance, and efficiently incorporates them into an end-to-end framework. In LPW, aside from runtime information, all other messages are autonomously generated by LLMs using few-shot prompting, without the need for annotated corpora or additional training.

A unique feature of LPW is the incorporation of the plan verification (block (c) in Figure 1) as the natural language intended solution for visible tests to derive the reliable program solution. LPW

initially produces a solution plan that decomposes complex programs into several tractable sub-problems (intermediate steps) (Cheng et al., 2023; Zelikman et al., 2023; Jiang et al., 2023). LPW then verifies the solution plan against visible tests to assess its correctness, known as plan verification. For a visible test, the verification includes a text-based, step-by-step analysis to derive the output for each intermediate step and the final output, ensuring that the final output is consistent with the visible test result. Additionally, each inferred intermediate output is reviewed by LLMs (block (d) in Figure 1) to maintain logical consistency throughout the verification.

Different from other approaches that exclude the solution plan entirely from the code generation (Chen et al., 2023b; Zhong et al., 2024), LPW incorporates the LLM-generated plan and its verification in the initial code development to clarify the programming logic. This approach ensures that the initial code closely aligns with the problem description, thus reducing the need for subsequent refinements. The plan verification encompasses comprehensive conditions and logical specifications for solving visible tests, eliminating potential misunderstandings before code generation. This is akin to *Test-Driven Development*, where human developers validate the intended solution with test cases (Beck, 2022). Furthermore, LPW consistently integrates plan verification in the subsequent refinements. In contrast to previous studies (Chen et al., 2023b; Zhong et al., 2024; Shinn et al., 2023) that query LLMs to infer errors in the generated code when it fails a visible test, LPW prompts LLMs to compare the expected output of each intermediate step for solving the failed visible test, as recorded in the plan verification, against the execution trace on the failed visible test to identify discrepancies and further produce an error analysis (block (j) in Figure 1). This approach is more straightforward and reduces uncertainty. These discrepancies assist LLMs in accurately locating bugs and identifying logic flaws in the code implementation, and generating detailed refinement suggestions, as documented in the error analysis. Then, the error analysis when integrated with the code explanation serves as feedback to refine the code in LPW, surpassing conventional scalar or vector rewards and thereby improving the efficiency and accuracy of the refinement process.

We further explore a sampling variant of LPW named as SLPW. SLPW leverages the Upper Confidence Bound (UCB) algorithm (Auer et al., 2002) to balance the exploration and exploitation in debugging multiple generated code samples for optimizing overall performance. We evaluate LPW and SLPW on four text-to-code generation benchmarks: HumanEval (Chen et al., 2021), MBPP (Austin et al., 2021), and their extended test case variants, HumanEval-ET and MBPP-ET (Dong et al., 2023a). We conduct experiments on the proprietary LLM GPT-3.5 (Achiam et al., 2023), and open-source LLMs, Llama-3 (Meta, 2024) and Phi-3 (Abdin et al., 2024). The Pass@1 accuracy (Chen et al., 2021) is reported. The experiment results demonstrate that LPW and SLPW consistently improve text-to-code generation performance across all benchmarks and LLM backbones. Compared to the state-of-the-art LLM debugger, LDB (Zhong et al., 2024), LPW improves Pass@1 accuracy by around 4% across all benchmarks with the GPT-3.5 backbone and achieves up to 16.4% improvement on MBPP when using Llama-3 as the backbone. SLPW shows an additional 1% improvement over LPW with GPT-3.5 and increases accuracy by up to 5.6% over LPW on MBPP with Phi-3. When tested with the advanced GPT-4o (OpenAI, 2024) backbone, LPW and SLPW maintain their advantages, and SLPW achieves new state-of-the-art performance across all benchmarks. Notably, on two challenging benchmarks, APPS (Hendrycks et al., 2021) and CodeContests (Li et al., 2022), LPW and SLPW improve Pass@1 accuracy by around 10% and 5%, respectively, compared to LDB with the GPT-4o backbone.

We outline the key contributions in this paper as follows:

- We introduce an end-to-end large language model programming workflow, LPW, that draws inspiration from conventional software development models while streamlining and tailoring them specifically for text-to-code generation. LPW significantly improves the code generation accuracy over the state-of-the-art methods.

- In LPW, we derive the intended solution for visible tests, represented by the plan verification, through querying LLMs to validate the correctness of the LLM-generated solution plan on visible tests before code implementation. The plan verification clarifies all conditions, flow logic, arithmetic operations, and punctuation specifications required to solve the visible tests for the given problem, thereby increasing the LLMs' confidence during both the initial program generation and subsequent debugging processes.

- We investigate SLPW, a sampling variant of LPW, and show that debugging across multiple program samples can further enhance performance and set new state-of-the-art results.

- We conduct extensive experiments across six text-to-code generation benchmarks to validate the performance of LPW and SLPW with various LLM backbones, provide a comprehensive analysis of their performance and failure cases, and highlight the existing challenges.

## 2 PROBLEM FORMULATION

We follow the problem formulation for text-to-code generation as outlined in Jiang et al. (2023), Chen et al. (2023b), and Zhong et al. (2024). The text-to-code generation problem is formulated as a triple $\mathcal{P} = \langle Q, T_v, T_h \rangle$, where $Q$ represents the problem specifications described in natural language, and $T_v$ and $T_h$ are sets of visible and hidden tests, each containing input-output pairs $(t^i, t^o) \in T = T_v \cup T_h$. The goal is to leverage the LLM $\mathcal{M}$ to generate a program function $f$, $\mathcal{M} \rightarrow f$, that maps each input $t^i$ to its corresponding output $t^o$ for all pairs in $T$, i.e., $f(t^i) = t^o$, for $(t^i, t^o) \in T$. We note that $T_h$ remains hidden during both solution generation and code implementation phases and only becomes visible if the generated $f$ passes $T_v$. In LPW, for all components shown in Figure 1, the problem description $Q$ is, by default, concatenated with task-specific prompts to produce the desired response from LLMs.

## 3 WORKFLOW STRUCTURE

In this section, we first detail the two phases of LPW separately and then elaborate on the iterative update strategies used in each phase.

**Solution Generation.** Figure 2 displays the overall workflow of the solution generation phase in LPW (part (a)), with an example programming problem for illustration (part (b)). LPW leverages the self-planning approach introduced by Jiang et al. (2023) to abstract and decompose the problem description $Q$ into a strategic and adaptable plan $\Pi$ at the start of the solution generation phase. For a problem in HumanEval described by block (1) in Figure 2, its example solution plan is illustrated at block (3). However, the LLM-generated plan $\Pi$ may occasionally be incorrect, misguiding subsequent program generation. To avoid this, LPW queries the LLM to verify $\Pi$ against all visible tests $T_v$. The LLM-responded plan verification $\mathcal{A}(\Pi, T_v)$ delivers a step-by-step anal-

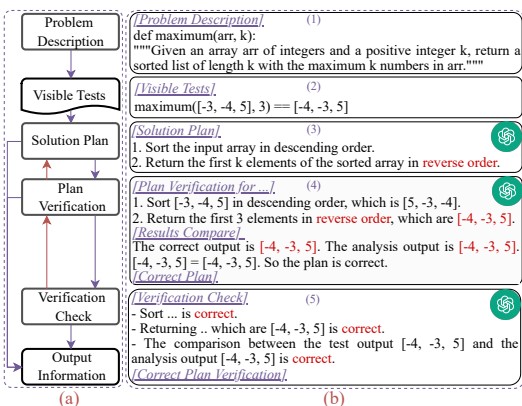

Figure 2: (a): An illustrated workflow of the solution generation phase in LPW. (b): Example message fragments corresponding to each workflow component for a HumanEval problem (*120th*) with the GPT-3.5 backbone. The detailed messages are available in Section 6.

ysis, including all intermediate results and final derived outputs for all visible tests $T_v$ based on $\Pi$. For each $t_v \in T_v$, its verification $\mathcal{A}(\Pi, \{t_v\})$ compares the derived output $t_v^{o'}$ with the ground-truth output $t_v^o$ to assess the correctness of $\Pi$, as outlined at block 4 in Figure 2. If $\Pi$ is successfully verified on all visible tests, where in $\mathcal{A}(\Pi, T_v)$, $t_v^{o'} = t_v^o, \forall t_v \in T_v$, then the plan verification $\mathcal{A}(\Pi, T_v)$ is reviewed by the LLM again to ensure the accuracy of all intermediate results, since each intermediate step result is used in locating bugs and providing refinement suggestions when compared with the code runtime information on the failed visible test. If all intermediate outputs in $\mathcal{A}(\Pi, T_v)$ are validated as correct by the LLM as shown at block 5 in Figure 2, $\mathcal{A}(\Pi, T_v)$ is treated as the intended solution for $T_v$. The plan $\Pi$ and its verification $\mathcal{A}(\Pi, T_v)$ serve as the output of the solution generation phase, guiding code development and refinements in the code implementation phase.

**Code Implementation.** Figure 3 shows the overall workflow of the code implementation phase in LPW (part (a)), using the same problem from Figure 2 as an illustration (part (b)). LPW develops an initial program $f$ by prompting the LLM with the problem description $Q$ (block (1) in Figure 2), along with plan $\Pi$ and its verification $\mathcal{A}(\Pi, T_v)$ from the solution generation phrase. Subsequently, LPW queries the LLM to add *print statements* for each line in $f$, resulting in $f_p$, and then executes $f_p$ on visible tests $T_v$. If $f_p$ successfully solves $T_v$, LPW validates it on the hidden tests $T_h$ to report Pass@1 accuracy. Otherwise, LPW collects the runtime information on the first failed visible test

$\bar{t_v}$, indicating that the implementation in $f$ deviates from the specifications in $\mathcal{A}(\Pi, \{\bar{t_v}\})$. Blocks 1-3 in part (b) of Figure 3 depict an initial program $f$ (block (1)) that fails on a visible test $\bar{t_v}$ (block (2)) and its execution trace (block (3)) on $\bar{t_v}$ after adding print statements. We omit $f_p$ from Figure 3 to keep the discussion concise. LPW instructs the LLM to conduct an error analysis by identifying inconsistencies between the intermediate outputs recorded in the execution trace of $\bar{t_v}$ and the expected outputs documented in the verification $\mathcal{A}(\Pi, \{\bar{t_v}\})$, analyzing the causes, and offering refinement suggestions (block (4)). Subsequently the error analysis and code explanation for $f$ generated by the LLM (block (5)) are concatenated as the prompt to generate the refined program $f'$ (block (6)). The code explanation helps the LLM align the text-based error analysis with the code implementation. LPW replaces $f$ with the refined program $f'$ and revalidates the updated $f$ against the visible tests $T_v$ to assess the necessity for further refinements.

**Iterative Updates**. LPW includes two update steps in the solution generation phase to enable *self-correction* as indicated by the red arrows in Figure 2: 1) when the plan verification inferred ultimate output differs from the ground-truth output for a visible test, where $t_v^{o'} \neq t_v^o, \exists t_v \in T_v$ in $\mathcal{A}(\Pi, T_v)$, a revised solution plan $\Pi'$ is included in the LLM response to substitute the original plan; 2) when the LLM detects any incorrect intermediate values in $\mathcal{A}(\Pi, T_v)$ e.g., contextual inconsistencies, mathematical miscalculations, or logical reasoning errors, LPW prompts the LLM to regenerate the plan verification. These update methods ensure that the solution plan $\Pi$ and its verification $\mathcal{A}(\Pi, T_v)$ achieve the necessary precision, as well-formed $\Pi$ and $\mathcal{A}(\Pi, T_v)$ are essential for the subsequent code generation accuracy (Jiang et al., 2023). In the code implementation phase, the code refinement process acts as an update mechanism,

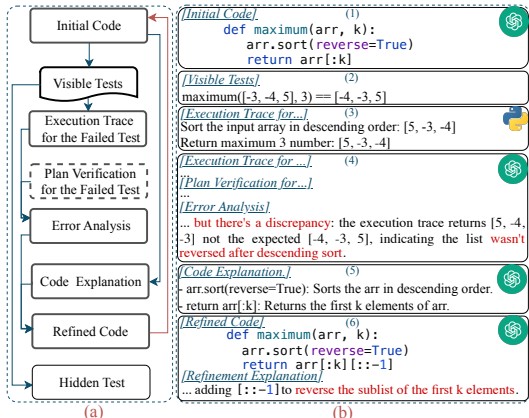

Figure 3: (a): An illustrated workflow of the code implementation phase in LPW. (b): Example message fragments extending from Figure 2 and corresponding to each workflow component. See Section 6 for detailed messages.

replacing the program $f$ with the refined program $f'$ when $f$ fails the visible test $T_v$ as highlighted by the red arrow in Figure 3. Overall, for a problem $\mathcal{P}$, LPW iteratively revises the generated plan $\Pi$ and its verification $\mathcal{A}(\Pi, T_v)$, in the solution generation phase, until $\mathcal{A}(\Pi, T_v)$ infers the correct outputs for all visible tests $T_v$ and no error intermediate outputs are present in $\mathcal{A}(\Pi, T_v)$. Otherwise, LPW reports a failure for $\mathcal{P}$ when reaching the maximum iterations. Similarly, in the code implementation phase, LPW iteratively refines the generated program $f$ if bugs exist. This process continues until a refined $f$ successfully solves $T_v$, followed by Pass@1 accuracy calculation on hidden tests $T_h$, or LPW reports a failure for $\mathcal{P}$ upon reaching the maximum iteration limit.

## 4 LPW WITH SAMPLING

Text-to-code generation benefits from both multiple sampling and debugging. These two approaches have evolved orthogonally. We propose a sampling variant of LPW, referred as SLPW. SLPW follows the same workflow and update mechanism as LPW but incorporates multiple plan samples $\{\Pi_1, \ldots \Pi_k\}$ and program samples $\{f_1, \ldots f_q\}$. SLPW generates $k$ plan samples at the beginning of the solution generation phase. For each iteration, SLPW leverages the UCB algorithm to competitively select a plan $\Pi$ with the highest upper confidence interval. Then, SLPW performs the same verification process as LPW for $\Pi$. When SLPW verifies $\Pi$ over each visible test and the verification fails on a visible test, it uses the number of visible tests where the plan verification derives an accurate final output as a reward to update the confidence interval of $\Pi$, and $\Pi$ is replaced with the revised plan $\Pi'$. Alternatively, when SLPW checks the correctness of intermediate outputs in the plan verification for each visible test and encounters erroneous values, it uses the number of visible tests where the plan verification contains correct intermediate outputs as a reward to update the confidence interval of $\Pi$. SLPW outputs the first $q$, where $q \leq k$, solution plans along with their verifications for the subsequent code implementation phase when solution plans and their verifications are confirmed as correct within the iteration threshold. Otherwise, SLPW provides $[0, q)$

| | | HumanEval | | | HumanEval-ET | | | MBPP | | | MBPP-ET | | |
|---|---|---|---|---|---|---|---|---|---|---|---|---|---|
| | | Acc ↑ | Δ ↑ | SD | Acc ↑ | Δ ↑ | SD | Acc ↑ | Δ ↑ | SD | Acc ↑ | Δ ↑ | SD |
| GPT-3.5 | Baseline | 74.4 | - | 0.8 | 66.5 | - | 1.3 | 67.4 | - | 0.5 | 52.8 | - | 0.3 |
| | SP | 77.4 | 3.1 | 0.8 | 69.5 | 3.1 | 0.8 | 69.2 | 1.8 | 0.4 | 52.4 | -0.4 | 0.2 |
| | SD | 81.1 | 6.7 | 1.0 | 72.0 | 5.5 | 1.0 | 71.2 | 3.8 | 0.3 | 56.0 | 3.2 | 0.1 |
| | LDB | 82.9 | 8.5 | 1.0 | 72.6 | 6.1 | 1.0 | 72.4 | 5.0 | 0.3 | 55.6 | 2.8 | 0.2 |
| | **LPW (ours)** | **89.0** | **14.6** | **0.8** | **77.4** | **11.0** | **0.8** | **76.0** | **8.6** | **0.2** | **57.6** | **4.8** | **0.1** |
| | **SLPW (ours)** | **89.6** | **15.2** | **0.6** | **77.4** | **11.0** | **0.6** | **77.2** | **9.8** | **0.3** | **58.2** | **5.4** | **0.2** |
| Llama-3 | Baseline | 73.2 | - | 1.0 | 61.0 | - | 1.0 | 44.0 | - | 1.0 | 35.4 | - | 1.0 |
| | SP | 78.0 | 4.9 | 2.0 | 65.2 | 4.3 | 1.0 | 48.6 | 4.6 | 1.4 | 38.4 | 3.0 | 1.4 |
| | SD | 81.7 | 8.5 | 1.3 | 68.3 | 7.3 | 0.8 | 63.6 | 19.6 | 1.2 | 50.0 | 14.6 | 1.3 |
| | LDB | 84.1 | 11.0 | 1.7 | 72.0 | 11.0 | 0.8 | 57.2 | 13.2 | 1.6 | 44.8 | 9.4 | 1.4 |
| | **LPW (ours)** | **88.4** | **15.2** | **1.6** | **76.2** | **15.2** | **1.2** | **73.6** | **29.6** | **1.3** | **56.4** | **21.0** | **1.2** |
| | **SLPW (ours)** | **89.0** | **15.9** | **1.6** | **76.2** | **15.2** | **1.3** | **75.0** | **31.0** | **1.2** | **57.2** | **21.8** | **1.0** |
| Phi-3 | Baseline | 36.0 | - | 1.0 | 32.3 | - | 1.0 | 39.0 | - | 1.3 | 33.2 | - | 1.4 |
| | SP | 40.9 | 4.9 | 1.4 | 34.8 | 2.4 | 0.9 | 46.4 | 7.4 | 1.4 | 37.6 | 4.4 | 1.4 |
| | SD | 51.2 | 15.2 | 1.2 | 45.7 | 13.4 | 1.0 | 45.8 | 6.8 | 1.2 | 36.6 | 3.4 | 1.2 |
| | LDB | 65.9 | 29.9 | 1.6 | 54.9 | 22.6 | 0.9 | 52.4 | 13.4 | 1.6 | 42.8 | 9.6 | 1.4 |
| | **LPW (ours)** | **76.8** | **40.9** | **1.3** | **62.8** | **30.5** | **1.2** | **64.0** | **25.0** | **1.2** | **48.4** | **15.2** | **1.2** |
| | **SLPW(ours)** | **81.1** | **45.1** | **1.2** | **67.1** | **34.8** | **1.2** | **69.6** | **30.6** | **1.4** | **52.2** | **19.0** | **1.2** |

Table 1: Comparisons of Baseline, Self-Planning (SP), Self-Debugging (+Expl) (SD), LDB, LPW and SLPW in terms of Pass@1 accuracy (Acc) and improvement (Δ) with respect to Baseline across benchmarks HumanEval, HumanEval-ET, MBPP, and MBPP-ET with LLMs GPT-3.5, Llama-3, and Phi-3. Acc and Δ are measured in percentages. The standard deviation (SD) is calculated and reported based on three runs. Results for LPW and SLPW are in bold, and the best results are highlighted in red.

solution plans and their verifications as the output after reaching the iteration threshold. In the code implementation phase, SLPW initially generates a program for each plan and its verification. If no initial program solves $T_v$, SLPW applies the UCB algorithm to optimize refinements across multiple programs. It selects a program $f$, refines it, and updates the confidence interval of $f$ based on the number of passed visible tests, until a refined program addresses $T_v$ and reports the Pass@1 accuracy on $T_h$. Otherwise, the process terminates with a failure upon reaching the iteration threshold. The algorithm details are available in the Appendix A.1.

## 5 EXPERIMENTS

**Benchmarks**. We evaluate LPW and SLPW on the well-established text-to-code benchmarks HumanEval, MBPP, HumanEval-ET, and MBPP-ET, where the given context outlines the intended functionality of the program to be synthesized. HumanEval-ET and MBPP-ET introduce approximately 100 additional hidden tests, covering numerous edge cases for each problem in HumanEval and MBPP, thus being regarded as more reliable benchmarks for code evaluation (Dong et al., 2023a) . In HumanEval and HumanEval-ET, we treat the test cases described in the task description as visible tests, typically 2-5 per task. For MBPP, we consider its test set that contains 500 problems with 3 hidden tests per problem. We set the first hidden test as the visible test and treat the other two as hidden, consistent with studies (Chen et al., 2023b; Zhong et al., 2024; Ni et al., 2023; Shi et al., 2022). MBPP-ET uses the same set of problems and visible tests for each problem as MBPP.

**Experimental Setup**. We compare LPW and SLPW with the representative code generation approaches *Self-Planning* (SP) (Jiang et al., 2023) , *Self-Debugging (+Expl)* (SD) (Chen et al., 2023b) and *Large Language Model Debugger* (LDB) (Zhong et al., 2024). SP relies solely on the LLM-generated solution plan to produce the program solution in a single attempt without refinements. SD uses a *rubber duck debugging* approach in LLMs, where LLMs are prompted to provide explanations of generated programs as feedback for debugging. LDB, a state-of-the-art LLM debugger, segments generated programs into blocks based on the control flow graph, which facilitates bug detection and the refinement of each program block using runtime execution information in LLMs. A detailed comparison between the baseline methods and our methods are summarized in Tables 14 and 15 in the Appendix. We generate the seed programs with the same prompts and parameters introduced

| | | HumanEval | | HumanEval-ET | | MBPP | | MBPP-ET | | APPS | | CodeContests | |
|---|---|---|---|---|---|---|---|---|---|---|---|---|---|
| | | Acc ↑ | SD | Acc ↑ | SD | Acc ↑ | SD | Acc ↑ | SD | Acc ↑ | SD | Acc ↑ | SD |
| GPT-4o | Baseline | 91.5 | 0.3 | 81.7 | 0.3 | 78.4 | 0.4 | 62.6 | 0.2 | 41.7 | 0.3 | 28.0 | 0.5 |
| | LDB | 92.1 | 0.0 | 81.7 | 0.0 | 82.4 | 0.3 | 65.4 | 0.0 | 53.2 | 0.3 | 29.3 | 0.3 |
| | **LPW (ours)** | **97.0** | **0.3** | **84.1** | **0.3** | **84.8** | **0.2** | **65.8** | **0.1** | **62.6** | **0.3** | **34.7** | **0.3** |
| | **SLPW (ours)** | **98.2** | **0.0** | **84.8** | **0.0** | **84.8** | **0.3** | **66.0** | **0.1** | **64.0** | **0.3** | **35.3** | **0.3** |

Table 2: Pass@1 accuracy for Baseline, LDB, LPW, and SLPW on the same benchmarks in Table 1, as well as APPS and CodeContests when using the LLM GPT-4o (2024-05-13) as the backbone. SD stands for the standard deviation.

by Chen et al. (2023b) for SD and LDB and label the performance of seed programs as Baseline. We note that SD and LDB only perform refinements on seed programs that fail the visible tests. We experiment with various LLMs with different parameter sizes, including GPT-3.5 (turbo-0125, $\geq$175B), Llama-3 (70B-Instruct), and Phi-3 (14B-Instruct) to evaluate performance and demonstrate that both LPW and SLPW are model-independent.

We use the Pass@1 accuracy as the evaluation metric. We apply 2-shot prompting in both LPW and SLPW, with maximum 12 iterations for the solution generation phase and the code implementation phase, respectively. Similarly, we set the maximum number of debugging iterations to 12 for SD and LDB. In SLPW, $k$ is configured as 6, and $q$ is set to 3. For instance, the solution generation phase initially produces 6 plan samples. Subsequently, first 3 solution plans along with their verifications are returned within 12 iterations, or $[0, 3)$ solution plans and their verifications are provided as the output upon completing 12 iterations. All following experiments adhere to these parameter settings unless otherwise specified. Empirical discussion on parameters is available in Appendix A.2.

**Main Results**. Table 1 presents the Pass@1 accuracy for Baseline, SP, SD, LDB, LPW, and SLPW, along with their respective improvements over Baseline. Overall, LPW and SLPW consistently outperform all competing methods across all benchmarks and with various LLM backbones, showcasing the effectiveness of the proposed workflow and demonstrating the model-independent benefit of LPW and SLPW. Compared to LDB, LPW improves Pass@1 accuracy by 6.1%, 4.9% 3.6%, and 2%, on HumanEval, HumanEval-ET, MBPP, and MBPP-ET, respectively, with the GPT-3.5 backbone and achieves up to 16.4% improvement on MBPP when using Llama-3 as the backbone. LPW achieves the same performance as SLPW on HumanEval-ET when leveraging GPT-3.5 and Llama-3 as the backbones. SLPW slightly surpasses LPW by around 1% across all benchmarks, and achieves the best accuracy: 89.6% for HumanEval, 77.4% for HumanEval-ET, 77.2% for MBPP and 58.2% for MBPP-ET with GPT-3.5. Moreover, when using Phi-3 as the backbone, SLPW shows the highest improvement up to 5.6% over LPW on MBPP and up to 45.1% over Baseline on HumanEval. Compared with HumanEval and MBPP, all approaches perform worse on HumanEval-ET and MBPP-ET across different LLM backbones as thorough edge cases are contained in the hidden tests. This result is consistent with previous work (Dong et al., 2023b; Lin et al., 2024; Mu et al., 2023). The detailed failure analysis is available in the Appendix A.4.

**Results on Advanced LLM with Competitive Benchmarks**. To further demonstrate the effectiveness of LPW and SLPW, we evaluate their performance against LDB on the same benchmarks presented in Table 1, as well as on two competitive benchmarks, APPS and CodeContests, using the advanced LLM GPT-4o as the backbone. For APPS and CodeContests, we use subsets of 139 and 150 problems, respectively. APPS and CodeContests are unstructured benchmarks where visible tests are intermingled with the problem statements and function signatures are excluded. To align input data structure across benchmarks, we instruct GPT-4o to derive the optimal function signature and identify visible tests for each problem prior to conducting experiments. The experiment results are shown in Table 2 and the Pass@1 accuracy is reported. Similarly, the performance of the seed programs for LDB is referred to as Baseline. LPW outperforms Baseline and LDB across all benchmarks, achieving the same 84.8% accuracy as SLPW on MBPP. SLPW further improves performance and establishes new state-of-the-art Pass@1 accuracy across all benchmarks, notably achieving 98.2% on HumanEval. The outstanding performance of SLPW indicates that sampling and debugging are mutually complementary in enhancing code generation performance. For APPS and CodeContests, LPW and SLPW achieve over 62% and 34% accuracy, respectively, surpassing LDB by around 10% and 5% accuracy, highlighting the advantages of LPW and SLPW in tackling challenging benchmarks. GPT-4o is considered as a more powerful LLM. Baseline achieves 91.5%

and 28% accuracy on HumanEval and CodeContests without debugging, while LDB shows a negligible improvement of only 0.6% and 1.3% compared to Baseline on these two benchmarks. This underscores the limitations of debugging with coarse feedback. In contrast, the intended solution with respective to visible tests represented by the plan verification allows LPW and SLPW to clarify issues before code generation and efficiently correct bugs overlooked by LLMs. Appendix A.7 discusses problems GPT-4o fails to address and structured examples from APPS and CodeContests. LPW and SLPW consume additional tokens to generate plan and plan verification. However, LPW and SLPW demonstrate cost efficiency on the challenging HumanEval and CodeContests benchmarks. For a detailed analysis, see Appendix A.8.

**Learning from Test**. We further investigate the impact of the number of visible tests on SD, LDB, LPW, and SLPW that use visible tests to refine code. We propose a variant of MBPP-ET, denoted as MBPP-ET-3. In MBPP-ET-3, each problem's visible tests are the three hidden tests from MBPP, while the hidden tests are the extended test cases introduced in MBPP-ET. In other words, each problem in MBPP-ET-3 contains two more visible tests than in MBPP-ET, thereby providing informative feedback for better bug identification and program refinement in LLMs. Table 3 compares the Pass@1 accuracy of SD, LDB, LPW and SLPW on the

|  | MBPP-ET ↑ | MBPP-ET-3 ↑ | Δ ↑ |
|---|---|---|---|
| SD | 56.0 | 59.2 | 3.2 |
| LDB | 55.6 | 57.6 | 2.0 |
| **LPW (ours)** | **57.6** | **62.0** | **4.4** |
| **SLPW (ours)** | **58.2** | **63.4** | **5.2** |

Table 3: The impact on Pass@1 accuracy with additional visible tests using the GPT-3.5 backbone. MBPP-ET-3 includes two more visible tests per problem than MBPP-ET. Δ represents the accuracy improvement of MBPP-ET-3 over MBPP-ET. Pass@1 accuracy and Δ are measured as percentages.

MBPP-ET-3 benchmark with the GPT-3.5 backbone. LPW and SLPW dominate in both accuracy and improvement. SLPW achieves the highest accuracy of 63.4% on MBPP-ET-3 and the largest improvement of 5.2% over MBPP-ET. LPW and SLPW exploit visible tests by producing the step-by-step solutions for each visible test to clarify initial code logic and inform subsequent refinements, demonstrating superior efficiency in utilizing visible tests among the evaluated methods.

**Performance Analysis**. Figure 4 evaluates the Pass@1 accuracy of LPW and SLPW when considering different numbers of code implementation iterations on the HumanEval benchmark when using GPT-3.5 as the backbone. For SD and LDB, we allocate the same number of debugging iterations. We note that all evaluated approaches start from iteration 0, representing the Pass@1 accuracy before debugging. Specifically, for SD and LDB, this reflects the seed program (Baseline) accuracy, while for LPW and SLPW, it indicates the accuracy after generating a program for each plan and its verification produced from the solution generation phase. In Figure 4, Baseline and SP are plotted as straight lines with 74.4% and 77.4% accuracy, respectively, due to no debugging in-

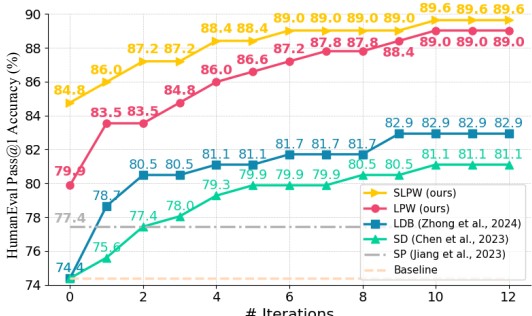

Figure 4: The impact on Pass@1 accuracy with the increased number of code implementation iterations/debugging iterations on the HumanEval benchmark when leveraging GPT-3.5 as the LLM backbone.

volved. Baseline and SP serve as the control group to illustrate when debugging methods surpass no-debugging methods. SD and LDB refine incorrect programs in Baseline, surpassing SP after two iterations. LPW starts debugging from an initial 79.9% accuracy, while SLPW begins from 84.8%. Both are higher than the 77.4% for SP, highlighting the importance of plan verification in initial code generation. LPW surpasses the best performance of SD and LDB after only one iteration, demonstrating its efficient code refinement strategy. The initial debugging accuracy of SLPW, 84.8%, exceeds the best performance of SD and LDB, showcasing the advantages of sampling. LPW and SLPW gradually refine the code and reach the highest Pass@1 accuracy by the 10th iteration.

**Ablation Study**. Table 4 shows the Pass@1 accuracy of different variants of LPW and SLPW on the HumanEval and MBPP benchmarks with GPT-3.5. The suffix -V denotes the exclusion of plan verification in both solution generation and code implementation phases; -S stands for the LPW variant that excludes the solution generation phase; while -C represents the removal of the code implementation phase, specifically omitting code refinements. For each problem, LPW-V

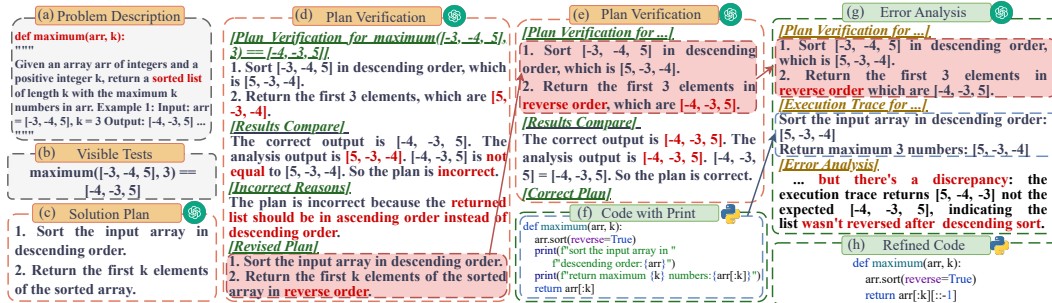

Figure 5: A case study of LPW on the *120th* problem in HumanEval, extending from Figures 2 and 3, using GPT-3.5. We omit certain components in Figures 2 and 3, e.g., the plan verification check and the initial code, and present incomplete prompts and responses to save space.

generates the initial program based on the unverified plan and repairs the program leveraging only code explanations and runtime information. LPW-S repairs the seed program that fails visible tests from Baseline, leveraging code explanations and runtime information but without plan and plan verification. LPW-C generates the program solution based on the plan and its verification without refinements. SLPW-S, SLPW-V, and SLPW-C maintain the same $k = 6$ and $q = 3$ settings as SLPW. SLPW-V generates $q$ programs, each derived from a corresponding unverified plan, and subsequently refines each program following the LPW-V framework. SLPW-S follows the same refinement approach as LPW-S to repair $q$ seed programs, generated in the same way as Baseline. SLPW-C employs the same strategy as LPW-C to create a program for each plan and its verification, but without applying refinements.

In Table 4, the performance decline of LPW-V and SLPW-V demonstrates the significance of plan verification. This confirms our hypothesis that plan verification serves as the intended solution for visible tests, improving the performance of LLMs in both initial code generation and subsequent refinements. Compared with LPW-S and SLPW-S, LPW-V and SLPW-V consider the unverified plan when drafting initial programs. However, the effect of the unverified plan is limited, as only the performance of LPW-V and SLPW-V on MBPP is improved compared with the results of LPW-S and SLPW-S. Besides, the removal of either phase in LPW or SLPW results in diminished performance, indicating that both the solution generation phase and the code implementation phase are crucial for optimal performance. For MBPP, both phases exhibit a similar impact in LPW and SLPW. In contrast, LPW-C experiences a significant 9.1% decrease on

| | HumanEval | | MBPP | |
|---|---|---|---|---|
| | Acc | Δ | Acc | Δ |
| LPW | 89.0 | - | 76.0 | - |
| LPW-V | 86.0 | -3.0 | 73.2 | -2.8 |
| LPW-S | 86.0 | -3.0 | 73.0 | -3.0 |
| LPW-C | 79.9 | -9.1 | 72.2 | -3.8 |
| SLPW | 89.6 | - | 77.2 | - |
| SLPW-V | 86.0 | -3.6 | 74.6 | -2.6 |
| SLPW-S | 86.0 | -3.6 | 74.4 | -2.8 |
| SLPW-C | 84.8 | -4.8 | 73.8 | -3.4 |

Table 4: Pass@1 accuracy (Acc) for different variants of LPW and SLPW with GPT-3.5. Δ denotes the decrease against LPW and SLPW. Acc and Δ are measured in percentages.

the HumanEval benchmark compared to LPW, as debugging plays a crucial role in reaching the accurate solution given the large number of visible tests in HumanEval. This also underscores the benefit of debugging for maintaining code quality when sampling is omitted. Meanwhile, the smaller 4.8% decrease for SLPW-C compared to SLPW on HumanEval demonstrates the advantage of sampling in the absence of debugging. See Appendix A.3 for additional ablation study.

# 6 CASE STUDY

Figure 5 illustrates example message fragments from LPW in the *120th* problem of HumanEval using the GPT-3.5 backbone. LPW successfully generates the correct program, while Baseline, SP, SD, and LDB all fail. This problem requires to return a sorted array with the maximum k numbers. However, in the problem description (block (a)), the unspecified order in the output array introduces ambiguity, confusing other methods. LPW struggles at the initial solution plan (block (c)), while the issue is clarified in the *[Revised Plan]*, during plan verification (block (d)). The visible test (block (b)) delineates the reverse order in the return array after sorting in descending order. The initial code with print statements (block (f)) fails on the visible test since the array is not reversed. Subsequently, its execution trace is compared with the plan verification (block (e)) to identify this bug, as described

in the *[Error Analysis]* in block (g). The refined code, which first sorts the array in descending order and then reverses the first $k$ elements into ascending order, successfully addresses this problem.

# 7 RELATED WORK

**Program Synthesis**. Program synthesis remains an open challenge of generating a program within a target domain-specific language (DSL) from given specifications. One prevalent approach involves searching the large space of possible programs. For example, generalized planning whose solution is formalized as a *planning program* with *pointers* (Segovia-Aguas et al., 2024; Lei et al., 2023) has demonstrated promising results in synthesizing program solutions for abstract visual reasoning tasks (Lei et al., 2024) when the DSL is carefully designed. However, hand-crafted DSLs often suffer from limited generalization capacity, and the huge search space diminishes its effectiveness. Recently, large language models trained on vast corpora have excelled in natural language processing (NLP) tasks and have been extended to code generation e.g., GPT-series (Achiam et al., 2023; OpenAI, 2024), Llama-series (Meta, 2024; Roziere et al., 2023; Touvron et al., 2023), and Claude-series (Anthropic, 2024). LPW and SLPW leverage the strengths of LLMs in NLP tasks to generate intended solutions in natural language. These text-based solutions demonstrate high-quality logical reasoning steps and satisfactory accuracy, thereby effectively aiding subsequent code generation.

**Prompting Techniques**. To imitate the logical chain in human brain when tackling reasoning tasks, prompting methods direct LLMs to decompose problems into solvable sub-problems (Jiang et al., 2023; Zhou et al., 2023; Lightman et al., 2024; Dhuliawala et al., 2023) and progressively infer the correct answer with intermediate outputs, as exemplified by chain-of-thought prompting (Wei et al., 2022; Kojima et al., 2022). Inspired by these studies, LPW and SLPW decompose a text-to-code problem into several sub-problems described by the solution plan and follow the chain-of-thought prompting idea to verify the solution plan against visible tests with step-by-step analysis. The generated plan and its verification provide step-by-step natural language instructions for code generation, aiding LLMs in both the initial code development and subsequent refinements.

**Code Refinement**. Accurate program solutions often require iterative refinements due to model limitations (Zhong et al., 2024; Chen et al., 2023b; Shinn et al., 2023). Various interactive approaches have been proposed to optimize debugging performance in LLMs, such as human feedback Chen et al. (2024); Le et al. (2022); Wu et al. (2023), trained models (Huang et al., 2023b; Le et al., 2022; Yasunaga & Liang, 2021), LLM-generated explanations (Chen et al., 2023b; Madaan et al., 2023; Shinn et al., 2023; Tang et al., 2023), and execution results (Zhong et al., 2024; Holt et al., 2024; Tian & Chen, 2023). Current state-of-the-art LLM debuggers, such as Self-Debugging and LDB, repair various seed programs to create the program solution. However, they encounter difficulties when the initial code substantially deviates from the original intent. Besides, without safeguarding, the refined code frequently diverges from the problem specifications. In contrast, LPW and SLPW develop initial code that adheres to the validated intended solution through plan verification, minimizing deviations from the problem description. The plan verification further guides code refinement, ensuring alignment with the problem specifications.

# 8 CONCLUSION

We propose LPW, a large language model programming workflow, for text-to-code generation tasks, which enables LLMs to accurately draft an initial program and effectively correct bugs. LPW uses various advanced code generation techniques and efficiently incorporates them into a two-phase development model. We further present SLPW, a sampling variant of LPW, where multiple initial programs are generated and then competitively refined as necessary. We evaluate LPW and SLPW on well-established text-to-code generation benchmarks across various LLMs. LPW significantly improves code generation accuracy compared to other existing approaches. SLPW achieves new state-of-the-art Pass@1 accuracy, with 98.2% on HumanEval, 84.8% on MBPP, 64.0% on APPS, and 35.3% on CodeContests benchmarks using GPT-4o as the backbone. These results highlight the effectiveness of our workflow in generating high-quality code and underscore the benefits of incorporating sampling and debugging. In the future, additional visible tests automatically generated by LLMs (Chen et al., 2023a) can be explored to improve the performance of LPW and SLPW.

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

# A APPENDIX

---

**Algorithm 1:** SLPW: Solution Generation Phase

---

**Input:** A problem description $Q$, large language mold backbone $\mathcal{M}$, visible tests $T_v$, maximum iterations for solution generation $I_s$, $k$ plan samples, $q$ output solution plans and their verifications.

**Output:** A set of plans with their verifications $S_{output} = \{(\Pi_1, \mathcal{A}(\Pi_1, T_v)), \dots, (\Pi_q, \mathcal{A}(\Pi_q, T_v))\}$.

1   // plan generation
2   $plans \leftarrow \mathcal{M}(Q, k, t = 0.4)$;   // generate $k$ initial $plans = \{\Pi_1, \dots \Pi_k\}$ with temperature $t = 0.4$
3   $S_{output} \leftarrow \emptyset$;
4   $i_s \leftarrow 0$;                  // set current iteration $i_s$ to 0
5   InitialUCB(len($plans$));         // initialize UCB algorithm with len($plans$) arms
6   **while** $i_s < I_s$ **do**
7      $\Pi \leftarrow$ SelectArm($plans$);       // select an arm $\Pi$ in $plans$ leveraging UCB algorithm
8      // plan verification generation
9      $\mathcal{A}(\Pi, T_v) \leftarrow \mathcal{M}(Q, \Pi, T_v)$;
10     $n, \Pi' \leftarrow \mathcal{A}(\Pi, T_v)$;    // number of visible tests $n$ where the plan verification derives an accurate final output, revised plan $\Pi'$
11     **if** $n = \text{len}(T_v)$ **then**
12        // verification intermediate output check
13        $z \leftarrow \mathcal{M}(Q, \mathcal{A}(\Pi, T_v), \Pi)$;    // number of visible tests $z$ where the plan verification contains correct intermediate outputs
14        **if** $z = \text{len}(T_v)$ **then**
15          $S_{output} \leftarrow S_{output} \cup (\Pi, \mathcal{A}(\Pi, T_v))$;      // save ($\Pi$, $\mathcal{A}(\Pi, T_v)$) if no error in $\mathcal{A}$
16          Delete($plans, \Pi$);           // delete arm $\Pi$ from $plans$ in UCB
17          **if** len($S_{output}$) $= q$ **then**
18            break;        // return first $q$ solution plans and their verifications
19        **else**
20          UpdateConfidence($\Pi, z$);    // update the confidence interval of $\Pi$ with reward $z$
21     **else**
22        $plans(\Pi) \leftarrow \Pi'$;              // replace $\Pi$ in $plans$ with revised plan $\Pi'$
23        UpdateConfidence($\Pi, n$);    // update the confidence interval of $\Pi$ with reward $n$
24     $i_s \leftarrow i_s + 1$;
25   **if** len($S_{output}$) $= 0$ **then**
26     **return** $\{\}$;    // return empty set if neither a valid plan nor its verification is created
27   **return** $S_{output}$;

---

## A.1 SLPW PSEUDO-CODE

Algorithms 1 and 2 present the pseudo-code for the solution generation phase and code implementation phase in SLPW. In Algorithm 1, SLPW initially generates $k$ plan samples, $plans$, on Line 2 with a temperature of $t = 0.4$. For all other LLM queries, $t$ is set to 0 by default to improve reproducibility. Lines 6-24 repeatedly select a plan $\Pi$ from $plans$ (Line 7), conduct verification (Line 9), and check the intermediate step outputs in the verification (Line 13) to generate $q$ solution plans along with their verifications as the output (Lines 14-18) when the plan verification and its intermediate outputs are successfully validated. In the solution generation phase, SLPW utilizes the UCB algorithm to select a plan $\Pi$ for further processing. SLPW treats each plan $\Pi$ as an arm and updates the confidence interval of $\Pi$ on Line 23 leveraging the number of visible tests $n$ where the plan verification derives an accurate final output, when verification fails on a visible test on Line 10, resulting in $n$ being smaller than the number of $T_v$ (Line 11). Additionally, SLPW updates the confidence interval of $\Pi$ on Line 20 using the number of visible tests $z$ where the plan verification contains accurate intermediate outputs, when erroneous intermediate values in the verification are detected on Line 13, leaning to $z$ being smaller than the number of $T_v$ (Line 14). The revised plan $\Pi'$ is empty (Line 10) when the plan verification confirms consistency between the derived outputs and the ground-truth outputs across all visible tests. We note that, unlike the standard *multi-armed bandit* problem where the distribution for each arm remains stable, SLPW replaces each plan $\Pi$ with the revised plan $\Pi'$ (Line 22) before updating confidence (Line 23) if the verification fails on a visible test and a revised plan $\Pi'$ is generated (Line 10). We hypothesize that $\Pi'$ remains closely related to $\Pi$, as typically only a few lines are changed. Therefore, the performance of $\Pi$ can offer valuable guidance for $\Pi'$.

---

**Algorithm 2:** SLPW: Code Implementation Phase

---

**Input:** A problem description $Q$, solution generation output $S_{output}$, large language mold backbone $\mathcal{M}$, visible tests $T_v$, maximum iterations for code implementation $I_c$.
**Output:** Final code $f$.

1  $programs \leftarrow \emptyset$;
2  $i_c \leftarrow 0$ ;                                                                  `// set current iteration i_c to 0`
3  `// initial code generation`
4  **for** plan $\Pi$, verification $\mathcal{A}(\Pi, T_v)$ in $S_{output}$ **do**
5  $\quad$ $f \leftarrow \mathcal{M}(Q, \Pi, \mathcal{A}(\Pi, T_v))$;  `// generate an initial program f for each Π and A(Π,T_v)`
6  $\quad$ $solved \leftarrow \text{Exe}(f, T_v)$ ;  `// return a boolean result solved after executing f on T_v`
7  $\quad$ **if** $solved$ **then**
8  $\quad\quad$ **return** $f$ ;                                               `// return f when f passes T_v`
9  $\quad$ $programs \leftarrow programs \cup (f, \mathcal{A}(\Pi, T_v))$ ;  `// record (f, A) for further refinements`
10  $\text{InitialUCB}(\text{len}(programs))$ ;  `// initialize UCB algorithm with len(programs) arms`
11  **while** $i_c < I_c$ **do**
12  $\quad$ $f, \mathcal{A}(\Pi, T_v) \leftarrow \text{SelectArm}(programs)$;  `// select an arm (f, A) using UCB algorithm`
13  $\quad$ $f_p \leftarrow \mathcal{M}(Q, f, \mathcal{A}(\Pi, T_v))$ ;  `// add print statements in f resulting in f_p`
14  $\quad$ $n, \bar{t_v}, \mathcal{T}(f, \bar{t_v}), \leftarrow \text{Exe}(f_p, T_v)$ ; `// the number of solved visible tests n, execution trace T`
$\quad\quad$ `on the first failed visible test t̄_v after executing f_p on T_v`
15  $\quad$ **if** $n = \text{len}(T_v)$ **then**
16  $\quad\quad$ **return** $f$ ;                                               `// return f when f passes T_v`
17  $\quad$ **else**
18  $\quad\quad$ `// error analysis`
19  $\quad\quad$ $\mathcal{O}(\mathcal{A}, \mathcal{T}) \leftarrow \mathcal{M}(Q, f, \mathcal{A}(\Pi, \bar{t_v}), \mathcal{T}(f, \bar{t_v}))$ ;  `// error analysis O by comparing A with T`
20  $\quad\quad$ `// code explanation`
21  $\quad\quad$ $\mathcal{E}(f) \leftarrow \mathcal{M}(Q, f)$ ;  `// generate explanation E for program f`
22  $\quad\quad$ `// code refinement`
23  $\quad\quad$ $f' \leftarrow \mathcal{M}(Q, f, \mathcal{E}(f), \mathcal{O}(\mathcal{A}, \mathcal{T}))$ ;  `// generate refined program f'`
24  $\quad\quad$ $programs((f, \mathcal{A}(\Pi, T_v))) \leftarrow f'$ ;  `// replace f in programs with refined program f'`
25  $\quad\quad$ $\text{UpdateConfidence}((f, \mathcal{A}(\Pi, T_v)), n)$ ; `// update confidence of (f, A(Π,T_v)) with reward n`
26  $\quad$ $i_c = i_c + 1$
27  **return** None ;  `// return none when no f passes T_v after reaching I_c`

---

| $q$ Iters | 1 | 2 | 3 | 4 | 5 |
|---|---|---|---|---|---|
| 12 | 89.0 | 89.0 | 89.6 | 89.6 | 89.0 |
| 20 | 89.0 | 89.6 | 90.2 | 90.2 | 89.6 |

Table 5: Pass@1 accuracy of SLPW on HumanEval with GPT-3.5 varies by iterations (Iters) in both the solution generation phase and the code implementation phase, as well as by the number of output plans along with verifications, $q$, in the solution generation phase.

Algorithm 2 summarizes the code implementation phase. During this phase, SLPW takes the output of the solution generation phase as input to generate a set of initial programs, $programs$, on Lines 4-9. Lines 11-26 repeatedly select a program $f$ (Line 12), add print statements to $f$ resulting in $f_p$ (Line 13), execute it on visible tests $T_v$ (Line 14), and return $f$ for further assessment on the hidden tests when $f_p$ solves visible tests $T_v$ (Lines 15-16). Otherwise, SLPW generates the refined program $f'$ (Line 23) with the error analysis $\mathcal{O}$ (Line 19) and the code explanation $\mathcal{E}$ (Line 21) when $f_p$ fails on $T_v$ and $\bar{t_v}$ is the first failed visible test (Line 14). The code implementation phase follows the same confidence interval update strategy as the solution generation phase due to the same hypothesize that the refine program $f'$ is closely related to $f$. It uses the UCB algorithm to competitively refine each program $f$ while replacing $f$ with the refined $f'$ (Line 24) and updating the confidence interval of $f$ with the number of solved visible tests $n$ (Line 25).

## A.2 PARAMETER STUDY

SLPW involves four hyper-parameters: (1) the number of iterations in the solution generation phase, (2) the number of iterations in the code implementation phase, (3) the number of plan samples $k$, and

| | Missing Conditions | Logic Errors | Differ from Intended Solution | No Code | Others |
|---|---|---|---|---|---|
| LPW | 33.3 | 0 | 5.6 | 50.0 | 11.1 |
| SLPW | 23.5 | 5.9 | 5.9 | 47.1 | 17.6 |

Table 7: The percentage of different failure reasons for LPW and SLPW on the HumanEval benchmark with GPT-3.5 as the backbone. *Missing Conditions* and *Logic Errors* arise from the same issues in the plan and plan verification. *Differ from Intended Solution* indicates the plan and plan verification are manually classified as correct, while the generated code deviates, resulting in failure. *No Code* represents the absence of valid plan and plan verification in the solution generation phase, leading to failure after reaching the maximum number of iterations. *Others* denotes error program solutions caused by various reasons that differ from the previously listed categories.

| | SD | LDB | LPW | SLPW |
|---|---|---|---|---|
| HuamEval | 22.6 | 28.6 | 44.4 | 47.1 |
| MBPP | 36.1 | 37.7 | 36.7 | 43.9 |

Table 8: The percentage of problems where Self-Debugging (+Expl) (SD), LDB, LPW, and SLPW generated programs solve the visible tests but fail the hidden tests, out of total failed problems for each method on HumanEval and MBPP, with GPT-3.5 as the backbone.

| | SD | LDB | LPW | SLPW |
|---|---|---|---|---|
| HuamEval | 4.3 | 4.9 | 4.9 | 4.9 |
| MBPP | 10.4 | 10.4 | 8.8 | 10.0 |

Table 9: The percentage of problems where Self-Debugging (+Expl) (SD), LDB, LPW, and SLPW generated programs pass the visible tests but fail the hidden tests, out of a total of 164 problems in HumanEval and 500 problems in MBPP, with GPT-3.5 as the backbone.

(4) the number of output plans along with verifications $q$ for further code implementation. We use the same iterations for the solution generation and code implementation phases to simplify analysis. To identify the optimal parameters for SLPW to achieve the best performance, we vary $q$ from 1 to 5, set the number of iterations to 12 and 20, and configure $k = 2 \times q$ to ensure that sufficient plan samples generate enough verifications for further code implementation. The results in Table 5 reveal that larger iterations and $q$ values generally improve performance on the HumanEval benchmark. However, an excessive number, $q = 5$, has a detrimental effect on performance compared to $q = 3$ and $q = 4$ with the same number of iterations. For the same $q$ settings, increasing the number of iterations tends to improve performance but consumes additional token resources. A larger $q$ value results in a greater number of initially developed programs in the code implementation phase, thereby raising the probability of passing visible tests. However, it also increases the risk of failing hidden tests due to less specific consideration on how to handle test cases during the initial program generation. Compared to $q = 3$, setting $q = 5$ results in a 2.4% improvement in instances where the generated program solves only the visible tests but fails the hidden tests, out of a total of 164 problems, with 12 iterations.

## A.3 ADDITIONAL ABLATION STUDY

Table 6 shows the performance of the variants of LPW and SLPW on the HumanEval and MBPP benchmarks using GPT-3.5 as the LLM backbone. The suffix -E denotes removing the code explanation in LPW and SLPW when generating the refined program in the code implementation phase. The code explanation facilitates LLMs in aligning text-based error analysis with code implementation when locating and refining incorrect program lines. It shows a greater impact in LPW compared to SLPW, as evidenced by the results of LPW-E and SLPW-E in Table 6.

| | HumanEval | | MBPP | |
|---|---|---|---|---|
| | Acc | Δ | Acc | Δ |
| LPW | 89.0 | - | 76.0 | - |
| LPW-E | 87.8 | -1.2 | 75.6 | -0.4 |
| SLPW | 89.6 | - | 77.2 | - |
| SLPW-E | 89.6 | 0 | 77.0 | -0.2 |

Table 6: Pass@1 accuracy (Acc) for the variations of LPW and SLPW with the GPT-3.5 backbone. Other metrics remain consistent with those in Table 4.

## A.4 ANALYSIS OF UNSOLVED PROBLEMS FOR GPT-3.5

Figure 7 compares the Pass@1 accuracy of LDB, LPW, and SLPW across different difficulty levels, *Easy*, *Medium*, and *Hard*, on the HumanEval benchmark using GPT-3.5. We apply the method de-

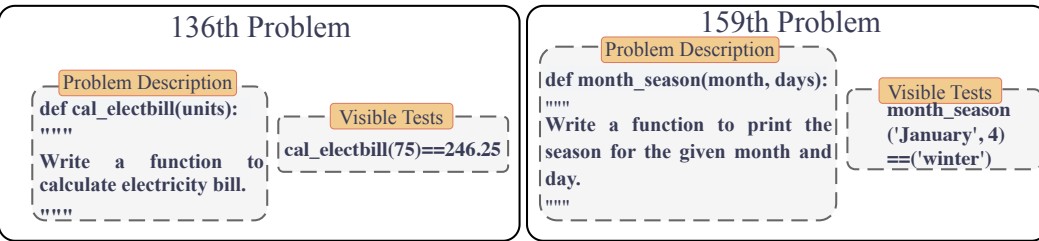

Figure 6: Example problems in MBPP.

scribed in Zhong et al. (2024) to generate the difficulty annotation in Figure 7 by querying GPT-3.5 with problem descriptions and canonical solutions. LPW and SLPW display convincing performance, exceeding 85% accuracy across all difficulty levels. For the *Hard* level, LPW and SLPW achieve 85.7% and 90.5% accuracy, in contrast to competing approaches whose performance notably degrades to below 70%.

LPW and SLPW achieve state-of-the-art performance among evaluated methods and show dominance compared to other LLM debuggers. We categorize the failure reasons for LPW and SLPW on HumanEval with GPT-3.5 into 5 types. Table 7 compares the percentage of different failure reasons out of the total unsolved problems for LPW and SLPW based on authors' manual review. Approximately half of the errors result from the *No Code* type, where the generated solution plan fails to be verified on the visible tests, or the resulting verification includes incorrect intermediate outputs in the solution generation phase, leading to failure after reaching the maximum iteration threshold. The second most common reason is *Missing Conditions*, originating from the same issues in

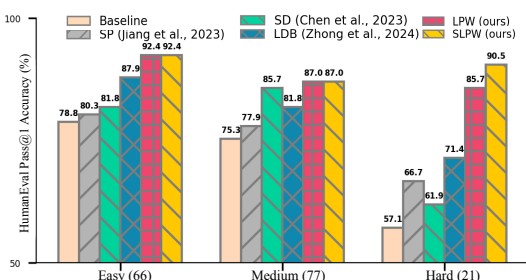

Figure 7: Pass@1 accuracy of Baseline, Self-Planning (SP), Self-Debugging (+Expl) (SD), LDB, LPW and SLPW across different difficulty levels, *Easy*, *Medium*, and *Hard* on the HumanEval benchmark when leveraging GPT-3.5 as the LLM backbone.

the plan and plan verification. Notably, LPW generated program solutions contain no logic errors, whereas SLPW produces only one program with a logic error. Both SLPW and LPW fail in the *91st* problem, where the generated programs are unable to solve the hidden tests due to deviations from the plan and plan verification (*Differ from Intended Solution*). The plan verification clearly specifies splitting the input string into sentences using delimiters ".", "?" or "!", but the generated code only handles the full stop case and ignores "?" and "!".

Tables 8 and 9 show the percentage of problems where Self-Debugging (+Expl) (SD), LDB, LPW, and SLPW generated program solutions pass the visible tests but fail the hidden tests, out of respectively failed problems and the total number of problems in the HumanEval and MBPP benchmarks using GPT-3.5 as the backbone. In Table 8, more than 40% of failures in LPW and SLPW result from solving the visible tests only on the HumanEval benchmark, since except for the *No Code* category, other reasons discussed in Table 7 could contribute to this issue. In contrast, less than 30% of problems in SD and LDB experience this issue on HumanEval as the larger number of failed problems in these two methods. In Table 8, all evaluated approaches show similar percentages on the MBPP benchmark, with the remaining failures arising from different reasons. We note that all methods tend to address visible tests only on the same set of problems in both the HumanEval and MBPP benchmarks, resulting in the similar percentage in each benchmark out of the total number of problems, as shown in Table 9. Meanwhile, all methods are prone to addressing visible tests only on MBPP rather than on HumanEval as indicated in Table 9. Compared to the detailed problem descriptions in HumanEval, the problem descriptions in MBPP are concise but lack clarity. For example, Figure 6 illustrates two problems in MBPP where LPW and SLPW generated solutions are tailored to the visible tests but deviate significantly from the canonical solution.

|  | Plan and Plan Verification | Correct Plan | Correct Plan Verification |
|---|---|---|---|
| LPW | 94.5 | 92.7 | 92.7 |
| SLPW | 95.1 | 93.9 | 93.3 |

Table 10: Percentage of problems where the LLM successfully generates the valid plans and plan verifications in the solution generation phase (first column); percentage of problems where the LLM-generated plans are manually classified as correct (middle column), considering no plan cases; and percentage of problems where the LLM-generated plan verifications are manually classified as correct (last column), considering no plan verification cases. SLPW generates multiple plans and plan verifications for each problem, and we consider them correct only when all are classified as correct. All percentages are reported using GPT-3.5 as the backbone on the HumanEval benchmark, with a total of 164 problems.

A.5   ACCURACY OF SOLUTION PLANS AND PLAN VERIFICATIONS USING GPT-3.5

We manually investigate the accuracy of solution plans and plan verifications generated by GPT-3.5 on the HumanEval benchmark, and the results are presented in Table 10. Overall, GPT-3.5 generates the correct solution plans and plan verifications in natural language for majority of problems. In LPW and SLPW, GPT-3.5 successfully produces plans and plan verifications for more than 94% of the problems. GPT-3.5 generates the correct plans for around 93% of the problems and achieves the similar accuracy for plan verifications. One common issue in the LLM-generated plan is the omission of certain conditions. For example, solution plan frequently overlooks uppercase situations and negative numbers. We note that the LLM-generated plan verification closely adheres to the solution plan. When the plan is accurate, the verification process strictly follows the plan logic, resulting in a correct analysis. Conversely, if the plan contains logical errors or omits edge cases, the verification process replicates these mistakes. Specifically, for LPW, all correct plans lead to accurate plan verifications, and vice versa. For SLPW, there is a single instance (*68th*) where the plan is correct, but the plan verification is classified as incorrect due to minimal logical flaws during inferring intermediate outputs.

We further manually explore the relationship between plan verification and program solution on the HumanEval benchmark with GPT-3.5. Table 11 evaluates the conditional probabilities between wrong code and wrong plan verification, as well as between correct code and correct plan verification. Typically, in LPW and SLPW, accurate plan verification significantly contributes to an accurate program solution, and vice versa. In LPW and SLPW, GPT-3.5 generates program solutions based on plans and plan verifications. Therefore, any accurate descriptions or mistakes, including missed conditions, in the plan and plan verification are propagated to the code. For SLPW, the *68th* problem's verification is classified as wrong, while the subsequently generated program is correct due to the sound underlying logic in the plan verification. This results in the value in the first column being less than 100%. When plan verifications are accurate, over 95% of program solutions are correct in LPW and SLPW. The remaining incorrect code instances result from unclear condition statements for hidden tests in plan verification, leading to an error program solution.

The results from Tables 10 and 11 highlight the impressive capabilities of LLMs in tackling text-to-code generation tasks when outputs are represented in natural language. Plan and plan verification generation accuracy is typically higher than code generation accuracy, underscoring the rationale behind LPW and SLPW, which produce the high-quality program solution by leveraging plan and plan verification. It is worth exploring methods to help LLMs overcome the challenges of translating natural language solutions into programs, given the strict lexical, grammatical, and semantic constraints. Additionally, exploring other types of natural language solution representations could improve code generation accuracy.

A.6   REFINEMENT CONSISTENCY IN LPW AND SLPW

LPW and SLPW allow multiple rounds of debugging to refine code based on error analysis generated by comparing the code execution trace and plan verification on the failed visible test. Additionally, LPW and SLPW query LLMs to generate refined code accompanied by a refinement explanation, detailing the modifications implemented to address the errors identified in the error analysis. For instance, Figures 12 and 5 illustrate two HumanEval problems where LPW successfully generates

| | Wrong Code ← Wrong Plan Verification | Correct Code ← Correct Plan Verification |
|---|---|---|
| LPW | 100 | 96.1 |
| SLPW | 90.9 | 95.4 |

Table 11: The relationship between LLM-generated code solutions and plan verifications on the HumanEval benchmark with GPT-3.5. The first column shows the percentage of problems where the LLM generates incorrect code solutions when plan verifications are incorrect; the second column shows the percentage of problems where correct code solutions are generated when plan verifications are correct. For SLPW, only the plan verification used to generate the final output code is manually evaluated.

the correct program through refinements informed by the error analysis using the GPT-3.5 backbone. We note that in LPW and SLPW, if the refined code is irrelevant to the error analysis, the entire debugging process degrades to a simple sampling approach, contradicting our original intent. As a result, we manually evaluate the debugging consistency among the generated error analysis (part (e)), the refined code (part (f)), and the refinement explanation (part (g)), as exampled in Figure 12. In LPW, only one refined code deviates from the error analysis yet still produces the correct solution, across all problems solved through debugging. SLPW achieves perfect consistency between the error analysis and the refined code. These results validate the effectiveness of the debugging steps in the code implementation phase for both LPW and SLPW, where the meaningful error analysis enables LLMs to produce the correct program with precise refinements.

### A.7 ANALYSIS OF UNSOLVED PROBLEMS FOR GPT-4O

#### A.7.1 HUMANEVAL

SLPW achieves 98.2% Pass@1 accuracy on HumanEval with the GPT-4o backbone, indicating only 3 unsolvable problems. We further investigate the reasons behind GPT-4o's failures on the *91st*, *132nd*, and *145th* problems as shown in Figures 13, 14, and 15, and attempt to generate the correct program solutions. The *91st* problem fails since GPT-4o misinterprets the linguistic distinction between the word and the letter; the *132nd* problem's ambiguous description challenges GPT-4o; and the incomplete description of the *145th* problem leads to failed plan verifications. GPT-4o successfully generates correct program solutions for 2 out of 3 problems, achieving 99.4% Pass@1 accuracy, by involving an additional visible test to validate the intended solution for the *91st* problem and providing a comprehensive problem description for the *145th* problem.

Figure 13 illustrates the *91st* problem in HumanEval, where the GPT-4o generated code (part (c)) contains an incorrect condition. The code verifies if the sentence starts with the letter "I", which is inconsistent with the problem description (part (a)) that requires the sentence to start with the word "I". The provided visible tests (part (b)) fail to clarify the correct condition, resulting in the error program passing the visible tests only. Inspired by the superior learning-from-test capacity discussed earlier, we convert a failed hidden test into a visible test, highlighted in red in part (d). Consequently, GPT-4o successfully generates the correct program, as shown in part (e).

Figure 14 displays the *145th* problem, where the incomplete problem description (part (a)) results in incorrect plan verification on visible tests (part (b)), leading to a failure after reaching the iteration threshold. The problem description requires returning a list sorted by the sum of digits but omits the specification regarding the sign of negative numbers. This omission confuses GPT-4o, resulting in consistently incorrect sorting when verifying the solution plan on the first visible test. We refine the problem description with a detailed explanation on handling both positive and negative numbers (part (c)), leading to the correct program solution, as shown in part (d).

Figure 15 illustrates the *132nd* problem, where ambiguity in the problem description (part (a)) challenges GPT-4o. The problem description lacks clarity on "a valid subsequence of brackets" and fails to specify the meaning of "one bracket in the subsequence is nested". We deduce the intended problem description by prompting GPT-4o with a canonical solution (part (d)). However, the LLM-generated description remains unclear and results in various erroneous programs. Furthermore, adding typically failed hidden tests to the visible tests (part (b)) is also ineffective in clarifying the correct logic. We acknowledge that a clearer description might contribute to the correct program.

|            | LDB  | LPW  | SLPW |
|------------|------|------|------|
| APPS       | 23.1 | 23.1 | 24.0 |
| CodeContests | 27.4 | 29.6 | 29.9 |

Table 12: The percentage of problems where LDB, LPW, and SLPW generated programs solve the visible tests but fail the hidden tests, out of total failed problems for each method in APPS and CodeContests, with GPT-4o as the backbone.

|            | LDB  | LPW  | SLPW |
|------------|------|------|------|
| APPS       | 10.8 | 8.6  | 8.6  |
| CodeContests | 19.3 | 19.3 | 18.7 |

Table 13: The percentage of problems where LDB, LPW, and SLPW generated programs pass the visible tests but fail the hidden tests, out of a total of 139 problems in APPS and 150 problems in CodeContests, with GPT-4o as the backbone.

However, some problems are inherently difficult to describe accurately in natural language without careful organization, posing challenges for LLMs.

### A.7.2 APPS AND CODECONTESTS

We preprocess problem descriptions in APPS and CodeContests, as shown in Figures 16 and 17, to maintain consistency in the input data structure. LPW and SLPW demonstrate significant improvements on APPS and CodeContests, exceeding around 10% and 5% Pass@1 accuracy, respectively, compared to LDB with GPT-4o. However, in contrast to their performance on the HumanEval and MBPP benchmarks, where LPW and SLPW achieve over 97% and 84% Pass@1 accuracy, the 62% accuracy on APPS and 34% accuracy on CodeContests indicate that even for the advanced LLM GPT-4o, code generation remains challenging when addressing complicated programming problems, such as those encountered in collegiate programming competitions like IOI and ACM (Hendrycks et al., 2021).

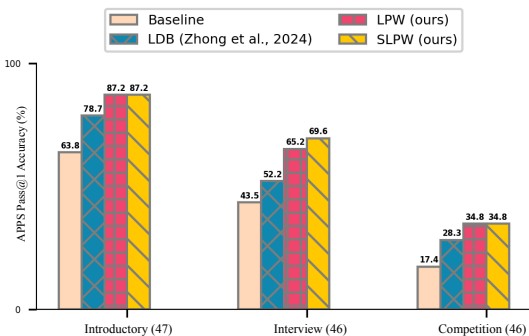

Figure 8: Pass@1 accuracy of Baseline, LDB, LPW and SLPW across different difficulty levels, *Introductory*, *Interview*, and *Competition*, on the APPS benchmark when using GPT-4o as the LLM backbone.

Figure 8 compares the Pass@1 accuracy of LDB, LPW, and SLPW across different difficulty levels, *Introductory*, *Interview*, and *Competition*, on the APPS benchmark using GPT-4o. LPW and SLPW consistently dominate in Pass@1 accuracy across all difficulty levels. LPW and SLPW show strong performance on the relatively easier levels, i.e., *Introductory* and *Interview*, surpassing LDB by around 9% and 15% accuracy, respectively, and outperforming Baseline by over 20% accuracy. For the problems belonging to the most challenging level, *Competition*, LPW and SLPW achieve 34.8% accuracy, compared to 28.3% for LDB and 17.4% for Baseline. However, all approaches experience a substantial decrease at the *Competition* level, emphasizing the need for further improvements.

Tables 12 and 13 present the percentage of problems where the generated program solutions from LDB, LPW, and SLPW solve visible tests but fail hidden tests out of the total failed problems and the total number of problems, respectively, on the APPS and CodeContests benchmarks using GPT-4o as the backbone. In Table 12, more than 20% of failures result from passing only the visible tests on the APPS benchmark, with this percentage increasing to around 30% on CodeContests for all evaluated methods. In Table 13, all approaches display similar percentages of solving visible tests only on each benchmark, ranging from around 10% on APPS to 19% on CodeContests. Compared to the results in Table 9, where LPW and SLPW address only visible tests in 5% and 10% of problems on the HumanEval and MBPP benchmarks, LPW and SLPW exhibit weaker performance on the more challenging APPS and CodeContests benchmarks. This is particularly evident on CodeContests, where the percentage is twice as high as APPS for LPW and SLPW. In APPS and CodeContests, each problem averages approximately 2 visible tests, while CodeContests includes more comprehensive hidden tests, averaging about 23 per problem compared to only around 5 per problem in APPS, increasing the likelihood of solving only the visible tests.

| | Analysis Before Coding | | Coding Without Debugging | Coding With Debugging | Sampling |
|---|---|---|---|---|---|
| | Planning | Plan Verification | | | |
| SP | ✓ | ✗ | ✓ | ✗ | ✗ |
| SD | ✗ | ✗ | ✗ | ✓ | ✗ |
| LDB | ✗ | ✗ | ✗ | ✓ | ✗ |
| LPW (ours) | ✓ | ✓ | ✗ | ✓ | ✗ |
| SLPW (ours) | ✓ | ✓ | ✗ | ✓ | ✓ |

Table 14: Features of Self-Planning (SP), Self-Debugging (+Expl) (SD), LDB, LPW, and SLPW with respect to code generation strategies.

## A.8 COST ANALYSIS

Figure 9 compares Pass@1 accuracy against the average token cost per program for LDB, LPW, and SLPW across four benchmarks using GPT-4o. LDB consumes fewer tokens per problem but achieves the lowest accuracy. LPW improves accuracy but requires additional token costs for generating and verifying the plan. SLPW achieves the highest accuracy but consumes the most tokens per problem due to the creation of multiple plans, plan verifications, and programs. When evaluated by the accuracy-to-token ratio, LDB achieves the highest efficiency on the simpler HumanEval and MBPP benchmarks, with accuracy gains of 5.85% and 1.04% per 1,000 tokens, respectively. In comparison, LPW achieves gains of 2.85% on HumanEval and 0.67% on MBPP, while SLPW achieves 1.42% on HumanEval and 0.45% on MBPP. On the challenging APPS benchmark, LDB uses fewer tokens per problem, while LPW and SLPW deliver significantly higher accuracy. As a result, LDB and SLPW yield the same accuracy gain of 0.39% per 1,000 tokens, whereas LPW demonstrates the highest efficiency at 0.43%. On the CodeContests benchmark, LDB, LPW, and SLPW exhibit similar token usage per problem, with LPW and SLPW achieving higher accuracy. Their accuracy gains, 0.17% for LPW and 0.16% for SLPW per 1,000 tokens, surpass LDB's 0.14%. LDB experiences low efficiency on APPS and CodeContests due to insufficient refinements, where multiple ineffective iterations consume significant token resources, yet the generated program remains flawed.

Figures 10 and 11 further illustrate the Pass@1 accuracy variations with token consumption for LDB, LPW, and SLPW on the APPS and CodeContests benchmarks using GPT-4o. LDB demonstrates a gradual and steady improvement in accuracy as increasesd token consumption. In contrast, LPW and SLPW start improving accuracy after a certain level of token consumption, due to the initial plan and plan verification generation. However, both LPW and SLPW subsequently show a sharp improvement in accuracy, ultimately surpassing LDB with fewer tokens consumed, highlighting the benefits of plan and plan verification in generating high-quality initial code and subsequent refinements. Challenging benchmarks aligns with LPW and SLPW usage scenarios, where the precise natural language solution described in the plan and plan verification is essential for logical completeness and understanding non-trivial bugs in the program, particularly when problems involve complex logical reasoning steps.

## A.9 LIMITATION

Like other debugging frameworks, LPW and SLPW are constrained by the imperfect reasoning abilities of LLMs. While the plans and plan verifications generated by LPW and SLPW show promising accuracy on current tasks, improving this accuracy could lead to better final code generation. Enhancing the reasoning capacity of LLMs remains an ongoing challenge. Additionally, both LPW and SLPW require a substantial number of tokens for plan generation and verification. Although the plan and verification have proven valuable on challenging benchmarks, reducing token usage is an important area for future enhancement. Furthermore, incorporating high-quality visible tests generated by LLMs could further improve the performance of our approach.

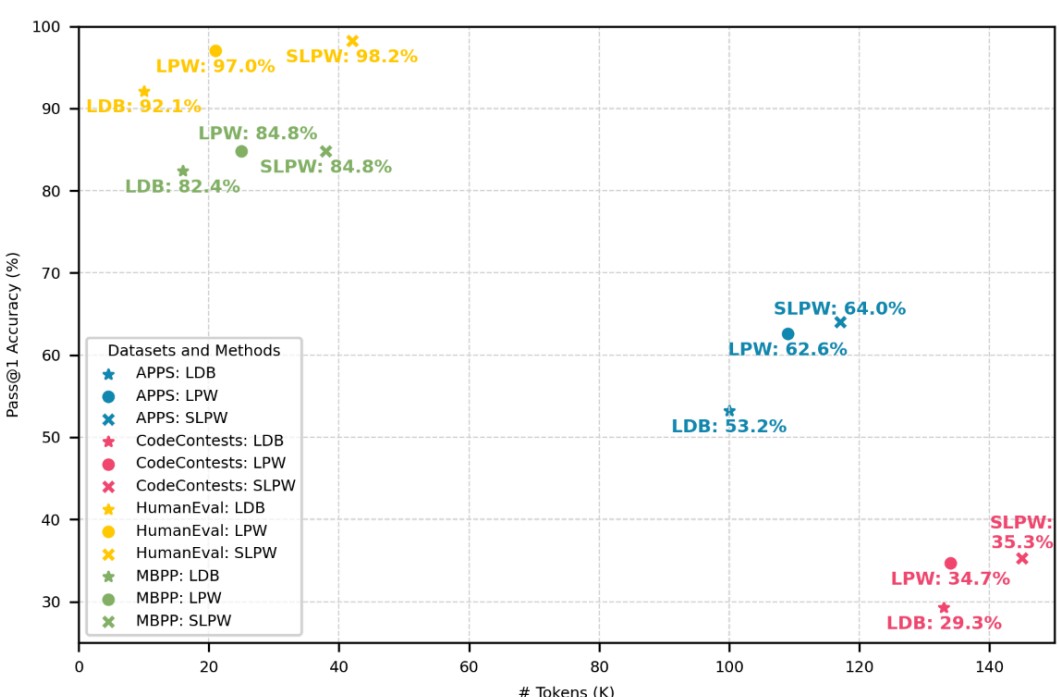

Figure 9: Pass@1 accuracy vs. average token cost per program for LDB, LPW, and SLPW on the HumanEval, MBPP, APPS, and CodeContests benchmarks using GPT-4o as the LLM backbone.

| | Code Explanation | Runtime Information | Natural Language Intended Solution |
|---|:---:|:---:|:---:|
| SD | ✓ | ✗ | ✗ |
| LDB | ✓ | ✓ | ✗ |
| LPW (ours) | ✓ | ✓ | ✓ |
| SLPW (ours) | ✓ | ✓ | ✓ |

Table 15: Features of Self-Debugging (+Expl) (SD), LDB, LPW, and SLPW with respect to code debugging approaches.

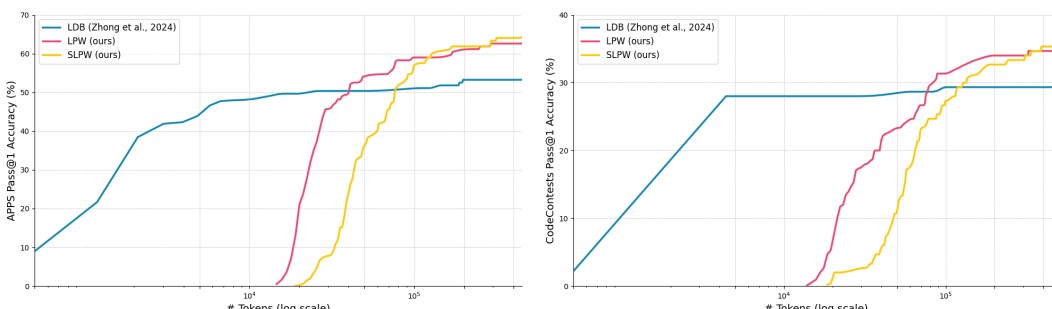

Figure 10: Pass@1 accuracy as a function of token consumption for LDB, LPW and SLPW on the APPS benchmark when using GPT-4o as the LLM backbone.

Figure 11: Pass@1 accuracy as a function of token consumption for LDB, LPW and SLPW on the CodeContests benchmark when using GPT-4o as the LLM backbone.

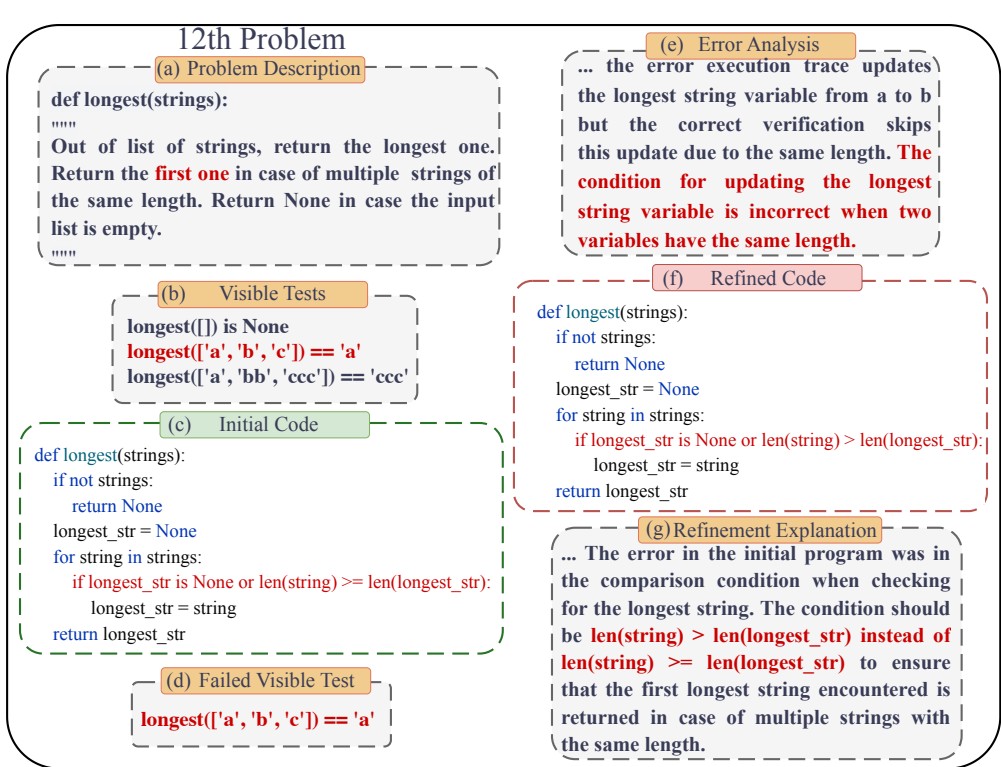

Figure 12: The *12th* problem in HumanEval, where LPW with GPT-3.5 generated initial code (part (c)) is unable to solve one of the visible tests (part (d)). The refined code (part (f)) successfully solves both visible and hidden tests based on the error analysis (part (e)). The modification in the refined code aligns with the error analysis, as evidenced by the refinement explanation (part (g)).

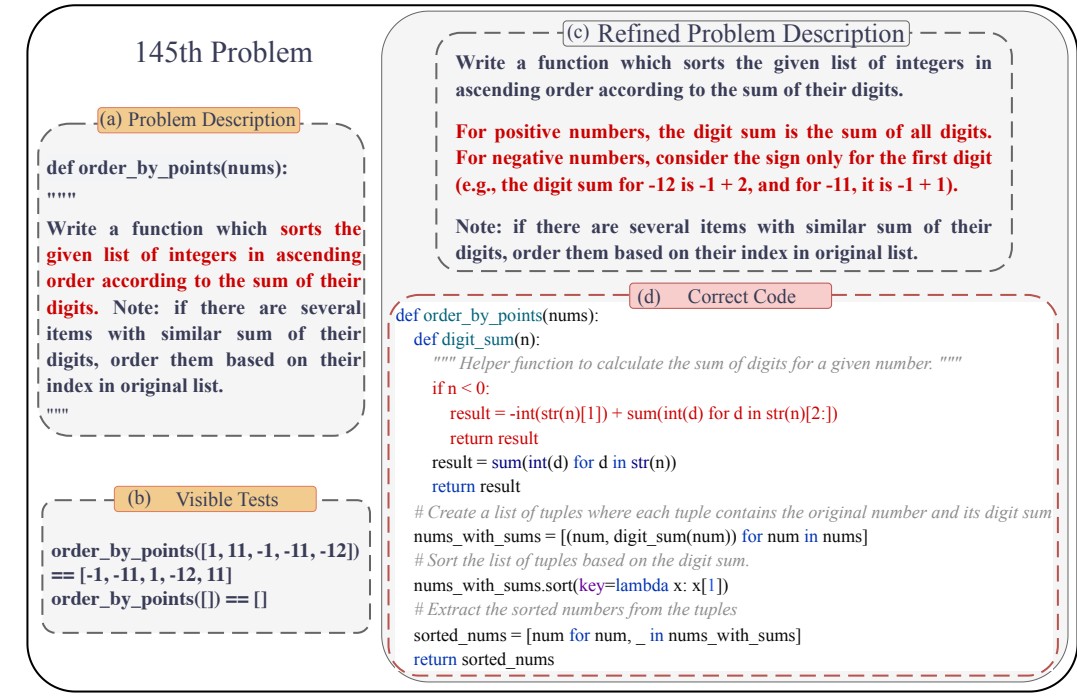

Figure 13: The problem description (part (a)) and visible tests (part (b)) of the *91st* problem in HumanEval, where GPT-4o generated code (part (c)) addresses the visible tests but fails the hidden tests. However, after converting a failed hidden test to a visible test (part (d)), GPT-4o successfully generates the correct program (part (e)).

Figure 14: The problem description (part (a)) and visible tests (part (b)) of the *145th* problem in HumanEval where GPT-4o fails to respond with a valid plan verification, resulting in failure. However, after refining the problem description (part (c)), GPT-4o successfully generates the correct program (part (d)).

## 132nd Problem

**(a) Problem Description**

```
def is_nested(string)
    """

    Create a function that takes a string as input
    which contains only square brackets. The
    function should return True if and only if there
    is a valid subsequence of brackets where at
    least one bracket in the subsequence is nested.
    """
```

**(b) Visible Tests**

```
is_nested('[[]]') == True
is_nested('[]]]]]]][[[[[]') == False
is_nested('[][]') == False
is_nested('[]') == False
is_nested('[[][]]') == True
is_nested('[[]][[') == True
```

**(c) Error Code**

```python
def is_nested(string: str) -> bool:
    counter = 0
    nested = False
    for char in string:
        if char == '[':
            # Increment counter for opening bracket
            counter += 1
            if counter > 1:
                # Set nested flag if there is more than one opening bracket
                nested = True
        elif char == ']':
            # Decrement counter for closing bracket
            counter -= 1
            if counter < 0:
                # Return False if counter goes negative (unbalanced brackets)
                return False
    # Return True if there was any nested bracket, else False
    return nested
```

**(d) Canonical Solution**

```python
def is_nested(string: str) -> bool:
    opening_bracket_index = []
    closing_bracket_index = []
    for i in range(len(string)):
        if string[i] == '[':
            opening_bracket_index.append(i)
        else:
            closing_bracket_index.append(i)
    closing_bracket_index.reverse()
    cnt = 0
    i = 0
    l = len(closing_bracket_index)
    for idx in opening_bracket_index:
        if i < l and idx < closing_bracket_index[i]:
            cnt += 1
            i += 1
    return cnt >= 2
```

Figure 15: The problem description (part (a)) and visible tests (part (b)) of the *132nd* problem in HumanEval, where the GPT-4o generated error code (part (c)) passes the visible tests yet fails the hidden tests. GPT-4o consistently generates incorrect programs despite providing additional visible tests or refining the problem description.

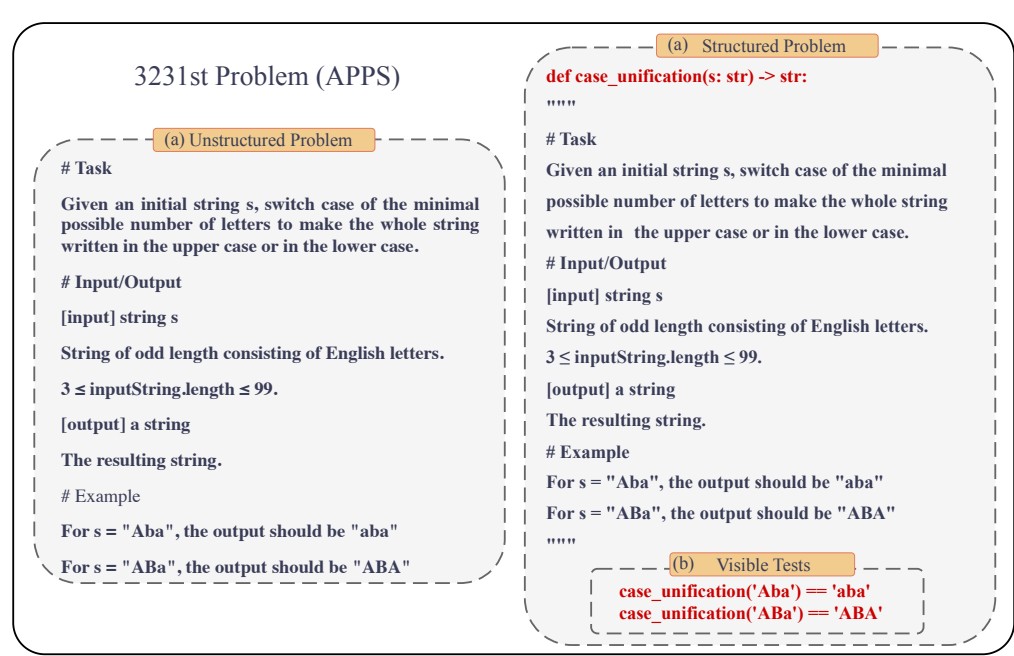

Figure 16: An example structured APPS problem with a function signature and visible tests, generated by instructing GPT-4o with the unstructured problem description.

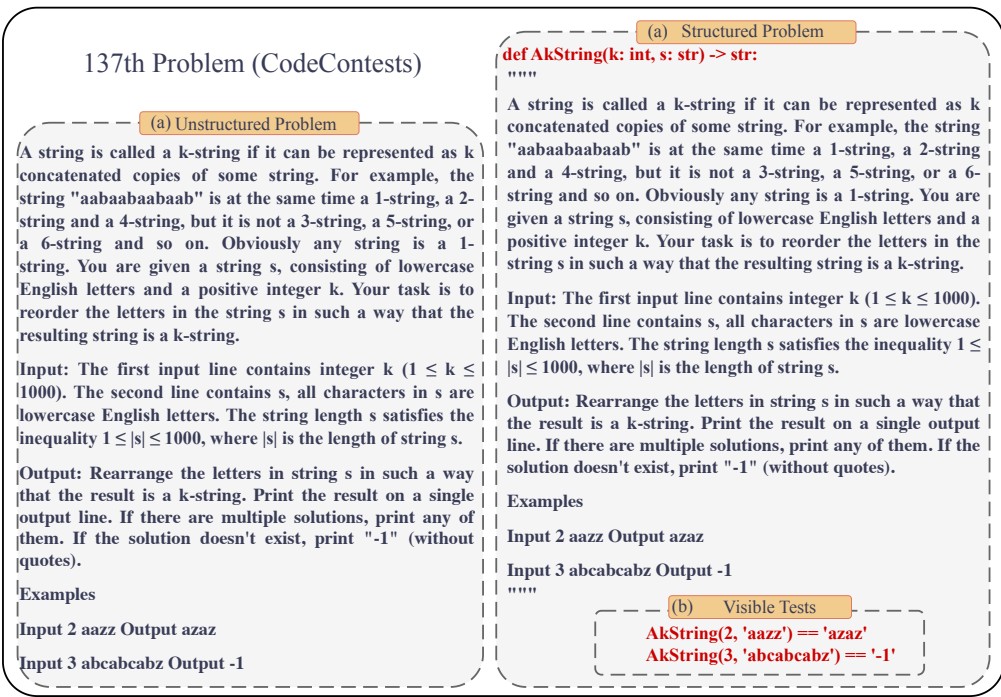

Figure 17: An example structured CodeContests problem with a function signature and visible tests, generated by instructing GPT-4o with the unstructured problem description.

## A.10 PROMPTS FOR LPW AND SLPW

We provide the LLM prompts used in LPW in Prompts 1 to 8. For conciseness, we only include one example in each prompt. Full prompts can be found in our released code.

Prompt 1: Prompt for plan generation

```
    You are a Python writing assistant that responds with a step-by-step
thought process (IN ENGLISH) to solve Python coding problems.

    You will be provided with a series of examples, where each example
begins with [Start Example] and ends with [End Example]. In each example,
 you will be presented with a Python coding problem, starting with [
Example Problem Description], which includes the function signature and
its accompanying docstring. You will then provide a reasonable solution
plan, starting with [Example Start Plan] and ending with [Example End
Plan], to solve the given problem.

[Start Example]
[Example Problem Description]
def encrypt(s):
    """
    Create a function encrypt that takes a string as an argument and
returns a string encrypted with the alphabet being rotated. The alphabet
should be rotated in a manner such that the letters shift down by two
multiplied to two places.
    """

[Example Start Plan]
Create an alphabet, biased by two places multiplied by two.
Loop through the input, find the letter biased by the alphabet.
Return the result.
[Example End Plan]
[End Example]

... Authors' notes: We omit another example for conciseness. The full
prompt can be found in our released code. ...

    Lastly, you will be given a Python writing problem, beginning with [
Problem Description], which includes the function signature, its
docstring, and any potential constraints. The phrase "Let's think step by
 step" will signal the start of the plan. Your task is to create a
solution plan, starting with [Start Plan] and ending with [End Plan].
```

Prompt 2: Prompt for plan verification

```
    You are a logical reasoner.

    You will be presented with several plan verification examples, each
starting with [Start Example] and ending with [End Example]. In each
example, you will be given a Python writing problem, starting with [
Example Problem Description], followed by the solution plan starting with
 [Example Solution Plan], and its verification process beginning with [
Example Plan Verification for X] for a test case X, starting with [
Example Test Cases]. During the verification process, intermediate
variables that need to be recorded are clearly identified at the outset,
starting with [Record Analysis]. Whenever the value of a recorded
intermediate variable is updated, the new result is clearly displayed,
```

```
beginning with [Record]. After the verification, the derived result is
compared to the correct test result, starting with [Results Compare]. If
the derived result matches the test result, the output will be [Correct
Plan]. If the derived result differs from the test result, the output
will be [Incorrect Plan], followed by the reasons for the discrepancy,
starting with [Incorrect Reasons], and the revised correct solution plan,
 beginning with [Start Revised Solution Plan] and ending with [End
Revised Solution Plan].

[Start Example]
[Example Problem Description]
def prime_number(n: int):
    """

    In range 0 to 100, returns n-th number that is a prime.
    """

[Example Solution Plan]
1. Iterate number through 0 to 100.
2. Check each number, if it's prime.
3. Keep track of the count of prime numbers found.
4. Stop when we find the n-th prime number.
5. Return the nth prime number.
[Example Test Cases]
assert prime_number(3)==5
[Example Plan Verification for assert prime_number(2)==3]
[Record Analysis]
The return value is the nth prime number, so all nth prime numbers need
to be clearly recorded!
1. Call the function prime_number(2).
2. According to line 1 in solution plan, Iterate number through 0 to 100.
3. According to line 2 in solution plan, Check if 0 is prime. It's not.
4. Move to next number 1.
5. According to line 2 in solution plan, Check if 1 is prime. It's not.
6. Move to next number 2.
7. According to line 2 in solution plan, Check if 2 is prime. It is a
prime.
8. According to line 3 in solution plan, the count of prime numbers is 1.
[Record]: 1st prime number is 2
9. Move to next number 3.
10. According to line 2 in solution plan, Check if 3 is prime. It is a
prime.
11. According to line 3 in solution plan, the count of prime numbers is
2.
[Record]: 2nd prime number is 3
12. According to line 4 in solution plan, Stop when we find the 2nd prime
 number.
13. According to line 5 in solution plan, Return the 2nd prime number,
which is 3.
[Results Compare]
The test correct output is 3. The logic analysis output is 3. 3=3. Thus,
the plan is verified to correctly handle all test cases.
[Correct Plan]
[End Example]

... Authors' notes: We omit another example for conciseness. The full
prompt can be found in our released code. ...

    Finally, you will be given a problem description, beginning with [
Problem Description], along with your generated solution plan, starting
```

with [Solution Plan], to solve the [Problem Description], and multiple
test cases starting with [Test Cases]. The phrase "Let's verify the plan"
 will indicate the beginning of the verification process, followed by
your verification steps to confirm whether your generated plan can pass
all test cases.

    For each test case, the verification must include [Record Analysis]
to track the intermediate variables at the beginning. If any intermediate
 variable value is updated during the reasoning process, the updated
value should be clearly displayed, starting with [Record]. Please include
 [Results Compare] to assess the derived outcome against the correct test
 output. If the derived result matches the test result, output [Correct
Plan] and proceed to the next test case. If the derived result does not
match the test result, output [Incorrect Plan], followed by the reasons
for the discrepancy, starting with [Incorrect Reasons]. Finally, provide
the revised solution plan, starting with [Start Revised Solution Plan]
and ending with [End Revised Solution Plan], to complete the process.

Prompt 3: Prompt for plan verification check

    You are a logical reasoner. Your goal is to identify any incorrect
logic within the logic verification process.

    You will be given several examples demonstrating how to evaluate a
logic verification process. Each example will begin with [Start Example]
and end with [End Example]. In each example, you will find the following:

[Example Problem Description] outlining the Python writing problem;

[Example Solution Plan] describing the approach to solve the problem;

[Example Plan Verification for X], applying the solution plan to a
specific test case X. In this process, the intermediate variables to be
tracked are analyzed at the start, marked by [Record Analysis]. Whenever
the value of a recorded intermediate variable is updated, its new value
is displayed starting with [Record]. The [Results Compare] section
compares the verification derived result with the correct test output;

[Example Verification Check for X], this section evaluates, step by step,
 whether the logic verification process for test case X is correct.

    If the verification is correct, the output will be [Correct Plan
Verification], and  please proceed to the next example. If the
verification is incorrect, explanation should be provided and [Incorrect
Plan Verification] will be the output to conclude the evaluation.

[Start Example]
[Example Problem Description]
def addOne(message: str):
    """
    You are given a large integer represented as an integer array digits,
 where each digits[i] is the ith digit of the integer. The digits are
ordered from most significant to least significant in left-to-right order
. The large integer does not contain any leading 0's. Increment the large
 integer by one and return the resulting array of digits.
    """

[Example Solution Plan]
1. Convert the list of digits into a number.

2. Increment the number by one.
3. Convert the incremented number back into a list of digits and return
it.

[Example Plan Verification for assert addOne([1,2,3])==[1,2,4]]
[Record analysis]
The return value is the incremental resulting array of digits, so the
incremental resulting array of digits needs to be clearly recorded!

According to line 1 in solution plan, convert [1,2,3] to the number 123.
According to line 2 in solution plan, Increment 123 by one to get 124.
According to line 3 in solution plan, convert 124 back into the list
[1,2,4]
[Record]: incremental resulting array is  [1,2,4]
According to line 3 in solution plan return incremental resulting array
[1,2,4].

[Results Compare]
The test correct output is [1,2,4]. The logic analysis output is
[1,2,4]. [1,2,4]=[1,2,4]. So the plan is verified to correctly handle all
 test cases.

[Correct Plan]

[Example Verification Check for assert ddOne([1,2,3])==[1,2,4]]:
"Convert [1,2,3] to the number 123" is correct!
"Increment 123 by one to get 124" is correct! since 123+1=124
"Convert 124 back into the list [1,2,4]" is correct!
"return incremental resulting array [1,2,4]" is correct!

In [Results Compare] "The test correct output = [1,2,4]" is correct! "The
 logic analysis output = [1,2,4]" is correct! The results comparison
"[1,2,4]=[1,2,4]" is correct!

All analysis steps are correct!

[Correct Plan Verification]

[Example Plan Verification for assert addOne([-1,2])==[-1,1]]
[Record analysis]
The return value is the incremental resulting array of digits, so the
incremental resulting array of digits needs to be clearly recorded!
According to line 1 in solution plan, convert [-1,2] to the number 12.
According to line 2 in solution plan, Increment 12 by one to get 13.
According to line 3 in solution plan, convert 13 back into the list [1,3]
[Record]: incremental resulting array is  [1,3]

According to line 3 in solution plan return incremental resulting array
[1,3].

[Results Compare]
The test correct output is [-1,1]. The logic analysis output is [-1,1].
[-1,1]=[-1,1]. So the plan is verified to correctly handle all test cases
.
[Correct Plan]

[Example Verification Check for assert addOne([-1,2])==[-1,1]]:

```
"Convert [-1,2] to the number 12" is incorrect. The analysis doesn't
correctly interpret the -1 and assumes all values are positive, the
sequence -1, 2 should form -12.
"Increment 12 by one to get 13" is correct, but as established, the
initial conversion should not yield 12.
"Convert 13 back into the list [1,3]" is correct!
"Return incremental resulting array [1,3]" is correct!

In [Results Compare] "The test correct output = [-1,1]" is correct! "The
logic analysis output = [-1,1]" is incorrect! The logic analysis result
is [1,3] mentioned in the verification "return incremental resulting
array [1,3]". The results comparsion "[-1,1]=[-1,1]" is incorrect! The
logic analysis result is [1,3] and [-1,1] is not equal [1,3].

The logic verification process for addOne([-1,2])==[-1,1] is incorrect.
The analysis doesn't correctly interpret the -1 and assumes all values
are positive, the sequence -1, 2 should form -12. The logic analysis
output = [-1,1] is incorrect! It is [1,3]. The results comparison is
incorrect since [-1,1] is not equal [1,3].

[Incorrect Plan Verification]

[End Example]

... Authors' notes: We omit another example for conciseness. The full
prompt can be found in our released code. ...

    Finally, you will be given a problem description, beginning with [
Problem Description], followed by your generated solution plan, starting
with [Solution Plan], to address the [Problem Description]. You will then
 work through multiple Plan Verification, each starting with [Plan
Verification for X], where X represents a test case. At the start of the
verification process, [Record Analysis] examines the intermediate
variables that should be tracked. During the logic verification, the tag
[Record] indicates any updates to the values of the recorded intermediate
 variables. The [Results Compare] section documents the comparison
between the  verification derived result and the expected test output.

    The phrase "Let's evaluate the verification" will indicate the start
of the evaluation for each verification process. This will be followed by
 your step-by-step verification check to assess whether each intermediate
 output in the verification process is correct, starting with [
Verification Check for X], as shown in the examples. If all intermediate
results in the verification process are correct, the output will be [
Correct Plan Verification], and you will proceed to the next verification
. If the verification process is incorrect, an explanation should be
provided, and [Incorrect Plan Verification] will be output to conclude
the evaluation.
```

Prompt 4: Prompt for initial code

```
    You are a Python writing assistant that only responds with Python
programs to solve a Python writing problem.

    You will receive several examples, each structured as follows,
beginning with [Start Example] and ending with [End Example]. Within each
 example, you will find a Python programming problem starting with [
Example Problem Description] and a solution plan starting with [Example
Solution Plan]. Additionally, you will receive plan verifications for
```

specific test cases. For each test case X, the plan verification is
labeled as [Example Plan Verification for X], providing a detailed
logical breakdown and variable value updates, which are recorded starting
 with [Record]. Following the verification, you will encounter the
example-generated program starting with [Example Generated Program]. The
program, marked from [Start Program] to [End Program], is generated based
 on the solution plan and plan verification, ensuring that the program's
execution aligns with the plan verification when test case X is used as
input.

[Start Example]

[Example Problem Description]
```
from typing import List
def get_closest_transition_character(word):
    """
    You are given a word. Your task is to find the closest transition
character from the right side of the word(case sensitive). The transition
 character is lowercase and the character after it is uppercase. If no
such lowercase character is found, return an empty string.
    >>> get_closest_transition_character("eAsy") == "s"
    """
```

[Example Solution Plan]
1. Reverse iterate through the characters of the word starting from the
last character from the right.
2. For each character, check if the current character is lowercase and if
 the character after it is uppercase.
3. If step 2 is satisfied,
4. return the current vowel character.
5. If no such vowel is found, return an empty string.

[Example Plan Verification for assert get_closest_transition_character("
eAsy")=="s"]

[Record analysis]
The return value is the closest transition character, so the closest
transition character should be recorded!

1. Call the function get_closest_vowel("eAsy").
2. According to line 1 in the solution plan, reverse iterate the word,
from the last character to the first character, so the last character is
"y"
3. According to line 2 in the solution plan, "y" is a lowercase but the
character after "y" is "s" and "s" is a lowercase.
4. Move to the next character based on the reverse iterate, so the
character is "s".
5. According to line 2 in the solution plan, "s" is a lowercase and the
character after 's' is 'A', and 'A' is uppercase.
6. According to line 3 in the solution plan, step 2 is satisfied,
7. [Record]: the closest transition character 's'
8. According to line 4 in the solution plan, return the current lowercase
 character 's'

[Example Generated Program]
[Start Program]

```
from typing import List
```

```
def get_closest_transition_character(word):
    """ You are given a word. Your task is to find the closest transition
 character from the right side of the word(case sensitive). The
transition character is lowercase and the character after it is uppercase
.
    >>> get_closest_transition_character("eAsy") == "s"
    """
    # reverse iterate the word
    for i in range (len(word)-1,-1,-1):
        current_character=word[i]
        if current_character.islower():
            if i!=0:
                after_character=word[i-1]
                if after_character.isupper():
                    return current_character
    return ""

[End Program]
[End Example]

... Authors' notes: We omit another example for conciseness. The full
prompt can be found in our released code. ...

    Finally, you will be provided with a Python writing problem, starting
 with [Problem Description]. A solution plan will follow, beginning with
[Solution Plan]. Next, you will receive several plan verifications. For
each test case X, the plan verification, starting with [Plan Verification
 for X] provides detailed logical reasoning steps to solve it.

    Once the plan verification is provided, the "Let's generate the
program" flag indicates the start of Python program generation. You will
then need to generate the Python program solution for the problem. The
plan verification serves as a constraint during program generation.  It
is essential to ensure that the execution of the generated program
remains consistent with [Plan Verification for X] when using test case X
as input. Additionally, the generated program should incorporate all
conditions noted in [Plan Verification for X] to solve test case X.
Please ONLY output the generated Python program, starting with [Start
Program] and ending with [End Program].
```

Prompt 5: Prompt for print statement generation

```
    You are a Python writing assistant that only responds with Python
programs with PRINT statements.

    You'll be provided with several examples structured as follows,
beginning with [Start Example] and ending with [End Example]. In each
example, you will be given a sample Python program, starting with [
Example Python Program]. You will also receive several plan verifications
 for specific test cases. For a test case X, its plan verification,
starting with [Example Plan Verification for X], includes a worded
description of the logic used to solve test case X. During the
verification, the intermediate variable that needs to be tracked is
clearly identified, starting with [Record Analysis] at the beginning, and
 any updates to its value are recorded, starting with [Record].

    Following this, you will be shown a Python program that includes
detailed print statements, starting with [Example Python Program with
Print Statements]. These print statements illustrate how the values of
```

the intermediate variables (described in the plan verification) are
modified during program execution, as well as how other variables in the
program change. These examples will guide you on where and how to add
print statements in your Python program.

[Start Example]

[Example Python Program]
```python
from typing import List
def get_closest_transition_character(word):
    """ You are given a word. Your task is to find the closest transition
 character from the right side of the word(case sensitive). The
transition character is lowercase and the character after it is uppercase
.
    >>> get_closest_transition_character("eAsy") == "s"
    """
    for i in range (len(word)-1,-1,-1):
        current_character=word[i]
        if current_character.islower():
            if i!=0:
                after_character=word[i-1]
                if after_character.isupper():
                    return current_character
    return ""
```

[Example Plan Verification for assert get_closest_transition_character("
eAsy")=="s"]
[Record analysis]
The return value is the closest transition character, so the closest
transition character should be recorded!

1. Call the function get_closest_vowel("eAsy").
2. According to line 1 in the solution plan, reverse iterate the word,
from the last character to the first character, so the last character is
"y"
3. According to line 2 in the solution plan, "y" is a lowercase but the
character after "y" is "s" and "s" is a lowercase.
4. Move to the next character based on the reverse iterate, so the
character is "s".
5. According to line 2 in the solution plan, "s" is a lowercase and the
character after 's' is 'A', and 'A' is uppercase.
6. According to line 3 in the solution plan, step 2 is satisfied,
7. [Record]: the closest transition character 's'
8. According to line 4 in the solution plan, return the current lowercase
 character 's'

[Example Python Program with Print Statements]
```python
from typing import List
def get_closest_transition_character(word):
    """ You are given a word. Your task is to find the closest transition
 character from the right side of the word(case sensitive). The
transition character is lowercase and the character after it is uppercase
.
    >>> get_closest_transition_character("eAsy") == "s"
    """

    print(f"Reverse iterate the word {word}")
    for i in range (len(word)-1,-1,-1):
        current_character=word[i]
```

```
        print(f"current character at index {i} is {word[i]}")
        if current_character.islower():
            print(f"current character {word[i]} is lowercase")
            if i!=0:
                print(f"There is a character after {word[i]}")
                after_character=word[i-1]
                print(f"character after {word[i]} is {word[i-1]}")
                if after_character.isupper():
                    print(f"character is {word[i-1]} is uppercase")
                    print(f"[Record]: the closest transition character {
word[i]}")
                    print(f"Return the closest transition character {word
[i]}")
                    return current_character

    print(f"no such lowercase character is found, return an empty string
")
    return ""
[End Example]

... Authors' notes: We omit another example for conciseness. The full
prompt can be found in our released code. ...

    Finally, you will be provided with a Python program, starting with [
Python Program], along with several plan verifications for specific test
cases. For each test case X, the plan verification, starting with [Plan
Verification for X], includes a detailed description of the logic used to
 solve test case X. In the plan verification, the intermediate variables
to be tracked are clearly analyzed at the beginning, starting with [
Record Analysis], and any updates to these variable values are recorded,
starting with [Record].

    The phrase "Let's add print statements" signals the start of the
process to incorporate print statements into the provided Python program.
 Your task is to add print statements that track how the variables in the
 program change. Ensure that the intermediate variable values (as
outlined in the plan verification) are printed using these statements.
Output your program with print statements, starting with [Start Program]
and ending with [End Program].
```

Prompt 6: Prompt for code explanation

```
    You are a Python interpreter.

    You will be given a Python program, and your task is to generate a
word-by-word explanation describing the effect of each line in the
program. You will be provided with several examples, each beginning with
[Start Example] and ending with [End Example]. In each example, you will
receive a Python programming problem, starting with [Example Problem
Description], along with an example Python program, marked as [Example
Python Program], which is generated to solve the given problem.
Additionally, you will be provided with an explanation of each line in
the example Python program, starting with [Example Explanation For Each
Line].

[Start Example]
[Example Problem Description]
def encode(message):
    """
```

```
    Write a function that takes a message, and encodes in such a way that
 replaces all letters in the message with the letter that appears 1 place
 after of that letter in the english alphabet and then delete letter is a
 vowel.
    The last letter z is execluded in the message. Assume the input only
containing letters.
    """

[Example Python Program]
def encode(message):
    encoded_message = ""
    for char in message:
        next_char = chr(ord(char) + 1)
        if next_char not in "aeiouAEIOU":
            encoded_message += next_char
    return encoded_message

[Example Explanation For Each Line]
The Python function encode is designed to manipulate a given string (
referred to as a message) by replacing the current letter with the letter
 that appears 1 place after it in the English alphabet and then skipping
the letter if it is a vowel:

Function Definition (def encode(message):): Defines a function named
encode that accepts one parameter, message. This parameter is intended to
 be a string that will be processed to create an encoded message.

Initialize Encoded Message (encoded_message = ""): Initializes a variable
 encoded_message as an empty string. This variable will store the encoded
 version of the message as characters are processed and approved for
inclusion.

For Loop (for char in message:): Iterates over each character in the
message. Each character is processed individually.

Calculate Next Character (replacechar = chr(ord(char) + 1)): For each
character in the message, this line calculates its next character that
appears 1 place after it in the English alphabet. It converts the
character to its ASCII value with ord(char), increments this value by 1,
and then converts it back to a character with chr().

Check if the resulting character is a Vowel (if replacechar in "
aeiouAEIOU":): Check if the resulting character (replacechar) after
incrementation is a vowel (either uppercase or lowercase is checked here)
. If it is a vowel, the continue statement is executed.

Add Character to Encoded Message (else: encoded_message += replacechar):
If replacechar is not a vowel, it is appended to encoded_message. This
builds up the final encoded string with the modified characters.

Return Encoded Message (return encoded_message): After processing all
characters in the original message, the function returns the fully
encoded string which consists of all non-vowel characters that are the
successors of the original characters in the ASCII sequence.
[End Example]

... Authors' notes: We omit another example for conciseness. The full
prompt can be found in our released code. ...
```

Finally, you will be presented with a problem description, starting
with [Problem Description], and your generated Python program, starting
with [Python Program], which is meant to solve the [Problem Description].
 After this, the "Let's generate the explanation" flag will signal the
beginning of the explanation phase. Your task is to generate a word-by-
word explanation for each line in the Python program, following the
format shown in the previous examples. Please skip the explanation for
any line that is a print statement. Output your explanation starting with
 [Start Explanation] and ending with [End Explanation].

Prompt 7: Prompt for error analysis

You are a logical reasoner. You will be provided with two logical
reasoning processes: [Plan Verification] and [Error Execution Trace].
Your task is to identify any errors in the [Error Execution Trace] by
comparing it with the [Plan Verification].

    You will be provided with several examples, each starting with [Start
 Example] and ending with [End Example]. In each example, you will
receive a Python programming problem, starting with [Example Problem
Description], along with an example of an incorrect Python program,
marked as [Example Error Program], generated for that problem. You will
also be provided with a detailed execution trace of the error program on
the failed test case X, labeled as [Example Error Execution Trace for X],
 including the intermediate variable values.

    Additionally, you will be provided with an example of the correct
logical reasoning process, labeled as [Example Plan Verification for X].
This process outlines the necessary steps to solve test case X accurately
, including condition checks and recording intermediate variable updates,
 starting with [Record]. Next, [Example Discrepancy Analysis] provides a
comparison between the Example Plan Verification and the Example Error
Execution Trace, highlighting output differences and identifying where
the Error Execution Trace deviates from correctness. Finally, [Example
Error Analysis] summarizes the errors identified in the [Example
Discrepancy Analysis] and proposes solutions to correct them.

[Start Example]
[Example Problem Description]
def is_palindrome(num):
    """
    check if a given integer is a palindrome.
    """

[Example Error Program]
def is_palindrome(num):
    num_str = str(abs(num))
    return num_str == num_str[::-1]

[Example Error Execution Trace for assert is_palindrome(-121)==False]
1. Convert the integer -121 to the string "121"
2. The integer string "121" is equal to the reversed string "121", the
result is True
3. Return True

[Example Plan Verification for assert is_palindrome(-121)==False]
[Record analysis]
The return value is the checking result about a given integer is a
palindrome, so the checking result should be clearly recorded!

```
1. Call the function is_palindrome(-121).
2. change integer to string, it is "-121"
3. check whether the string "-121" is equal to its reversed string
"121-", the checking result is False
4. [Record]: checking result = False
5. Return checking result False

[Example Discrepancy Analysis]
In the plan verification, the recorded value is the checking result:

Let's trace the "checking result" value in the plan verification when it
is first-time recorded (SKIP INITIALIZATION).

In the plan verification, the value of checking result is first-time
recorded in Line 4 after executing lines:
1. Call the function is_palindrome(-121).
2. change to integer to the string, it is "-121"
3. check whether the string "-121" is equal to its reversed string
"121-", the checking result is False
4. [Record]: checking result = False

In the  plan verification, the first-time update changes the checking
result value to False.

Let's trace the "checking result" value in the Error Execution Trace.
In Error Execution Trace, the value of checking result is first-time
recorded in Line 2 after executing lines
1. Convert the integer -121 to the string "121"
2. The integer string "121" is equal to the reversed string "121", the
result is True

In Error Execution Trace, the first-time update changes the checking
result value to True.

The checking result value in the  plan verification and Error Execution
Trace are NOT the same, due to False NOT equaling True when the checking
result value is first updated.

Let's carefully analyse the reason with step-by-step thinking:
In lines 1-4 in the  plan verification, the integer -121 is first
converted to the string "-121". Then "-121" is compared with its reversed
 string "121-". "-121" is NOT equaling "121-" so the result is False

In lines 1-2 in Error Execution Trace, the integer -121 is first
converted to the string "121". This is different from the plan
verification where converting -121 to string is "-121" rather than "121".
 Then "121" is compared with its reversed string "121". "121" is equaling
 "121" so the result is True.

[Example Error Analysis]
The error execution trace incorrectly converts the negative integer to
its negative integer string. The negative signal is missed. For example,
negative integer -121 should be converted to string "-121" but not "121.
To fix this error, the negative number must be considered and its
negative sign should be contained when converted to string. Such as
negative integer -121 should be converted to string "-121".
```

```
[End Example]

... Authors' notes: We omit another example for conciseness. The full
prompt can be found in our released code. ...

    Finally, you will be presented with a problem description, starting
with [Problem Description], along with your generated error program,
starting with [Error Program], which attempts to solve the [Problem
Description]. You will also receive a detailed execution trace, including
 intermediate variable values, for the failed test case X, starting with
[Error Execution Trace for X]. This trace is generated by the error
program. Additionally, you will be provided with a correct logical
reasoning process, labeled as [Plan Verification for X], which outlines
the necessary steps to solve test case X accurately, including condition
checks and recording intermediate variable updates, starting with [Record
].

    Following this, the "Let's do analysis" flag will indicate the start
of the analysis phase. Your task is to analyze where the [Error Execution
 Trace for X] deviates from the [Plan Verification for X], as
demonstrated in the examples. This analysis should be output starting
with [Discrepancy Analysis].  Finally, you should provide a summary of
the errors identified in the [Discrepancy Analysis], including the
reasons for these mistakes (IN ENGLISH) and suggestions on how to correct
 them, starting with [Error Analysis].
```

Prompt 8: Prompt for code refinement

```
    You are a Python program fixer. You need to correct an error Python
program based on the provided information.

    You will receive several examples, each structured as follows,
starting with [Start Example] and ending with [End Example]. Within each
example, you will find a Python programming problem, beginning with [
Example Problem Description], followed by an error program provided under
 [Example Error Program] for the given problem. You will then receive an
explanation for the error program, including a line-by-line explanation
starting with [Example Error Program Explanation].

    Additionally, an error analysis will be provided, starting with [
Example Error Analysis], describing the issues in the error program and
offering suggestions for refinement. You will then be provided with the
refined Python program under [Example Refined Program], based on the
error analysis. Following that, a refinement explanation, starting with [
Example Refinement Explanation], will be given to show which lines of the
 program were changed and explain the reasons for those changes.

[Start Example]

[Example Problem Description]
def is_palindrome(num):
    """
    check if a given integer is a palindrome.
    """

[Example Error Program]
def is_palindrome(num):
    num_str = str(abs(num))
    return num_str == num_str[::-1]
```

```
[Example Error Program Explanation]
Function Definition (def is_palindrome(num):): This line defines a
function named is_palindrome that takes one parameter, num. This
parameter is expected to be an integer.

Convert Number to Absolute String (num_str = str(abs(num))): A variable
num_str is initialized with the absolute value of num converted to a
string. The abs() function removes the sign from num if it's negative,
ensuring the palindrome check is based solely on the digits.

Check Palindrome and Return (return num_str == num_str[::-1]): This line
checks if the string representation of num_str is the same forwards and
backwards. It uses the slicing technique [::-1] to reverse the string. If
 num_str is equal to its reversed version, the function returns True,
indicating the number is a palindrome. Otherwise, it returns False.

[Example Error Analysis]
The error execution trace incorrectly converts the negative integer to
its negative integer string. The negative signal is missed. For example,
negative integer -121 should be converted to string "-121" but not "121.
To fix this error, the negative number must be considered and its
negative sign should be contained when converted to string.

[Example Refined Program]
def is_palindrome(num):
    num_str = str(num)
    return num_str == num_str[::-1]

[Example Refinement Explanation]
Program line (num_str = str(abs(num))) is changed to (str(num)) to
convert the negative integer to its negative integer string by deleting
the abs function to keep the negative representation as mentioned in the
the error analysis. (str(num)) can correctly convert negative integer
-121 to string "-121".

[End Example]

... Authors' notes: We omit another example for conciseness. The full
prompt can be found in our released code. ...

    You will be presented with a Python writing problem, starting with [
Problem Description]. The error program will be provided under [Error
Program], followed by an explanation of each line, starting with [Error
Program Explanation]. You will then receive an error analysis, starting
with [Error Analysis], which describes the issues in the error program
and provides refinement suggestions.

    The repair process will begin with the phrase "Let's correct the
program." Based on the error analysis, generate the refined program.
Output your refined program, starting with [Start Refined Program] and
ending with [End Refined Program], ensuring that ONLY the Python code is
included between these markers. Finally, provide a refinement explanation
, starting with [Refinement Explanation], detailing how the program was
modified to align with the error analysis.
```

