# OpenReview forum: "Planning-Driven Programming: A Large Language Model Programming Workflow"
_ICLR.cc/2025/Conference — ICLR 2025 Conference Withdrawn Submission_

### Official Review · Reviewer_T8rS · 2024-10-25

**Soundness:** 2
**Presentation:** 3
**Contribution:** 2
**Rating:** 3
**Confidence:** 5

**Summary:**

This paper presents a planning driven approach for the code generation problem. The exact setting is to generate code for a completely specified task, where some number of “visible” test cases are provided. The main idea in the paper is to use a planning approach driven by human software engineering practices, adding a design phase before the actual code generation phase. The design consists of a set of high-level steps with well-defined input-output behavior, and the code generation phase generates code for each high-level step.

The key insight here is that using a design verification step, where the execution of the designed solution is simulated to ensure that the plan is sound. Errors can be caught at an early stage during the design phase if the simulated execution on the visible test cases does not produce the expected results. The authors present two versions of their technique – one where each phase generates one single option, and the other where a sampling approach is used over to explore multiple possibilities for both the design plan and the implementation.

The authors compare their solution to one baseline (LDW) and show improvements over it for a number of models of varying capabilities including GPT-3.5, Llama-3, Phi-3, and GPT-4o.

**Strengths:**

+ Well-motivated technique

+ Improved results of baselines

**Weaknesses:**

- Choice of baselines: I am a little baffled by the choice of baselines. The techniques compared against are mostly targeted at debugging, not code generation. Why not pick a few of a plethora of the other code generation techniques: AgentCoder, MapCoder, CodeChain, WizardCoder, etc? If there is a specific reason why this is being done, it needs to be discussed in detail in the paper.

- Choice of benchmarks: The main evaluation is done on a very simple set of benchmarks. The MBPP and HumanEval datasets are more or less “solved”, and cannot be the basis for evaluating a planning driven technique. I would suggest that the authors do the deeper analysis on the APPS and CodeContests dataset, as well as consider what fragments of other datasets such as swebench will be applicable in the current setting.

- Flawed evaluation: Even with the chosen baselines and benchmarks,
** The improvements reported in the paper are in the order of a few percentage points. However, LLM-driven techniques, especially complex iterative ones with multi-step planning and reasoning, often have variance similar or larger than this. It is very hard to evaluate the gains without any reporting of variance or standard deviations.
** Why aren’t SD and SP used for the advanced APPS and CodeContests evaluations?
** The plots in figure 4 are very misleading -- having a non-zero value as the origin is one of the hallmarks of misleading reporting. At a glance, it makes the empirical improvements seem much better than they are.
** No reporting of cost-to-performance analysis. Does the improved performance of the current techniques come with an increased cost? If so, how much is this increased cost?

- Related work: There is insufficient discussion of related work, both specifically about code generation as well as general software engineering tasks. For example, there is a large body of recent work targeting the SWE-bench dataset that is arguably harder than the datasets chosen for comparison. There are planning works targeted at both targeted coding tasks and repository-level tasks (Bairi et al. FSE 2024, Zhang et al. ICLR 2023, etc) which are not discussed.

- Comparison to human software-engineering processes: The technique is presented as an analogue of waterfall and scrum software engineering processes. However, these processes are much more about human-centric planning. For example, scrum specifically prescribes the ideas of sprints, the idea of decision-making by lowest level actors, planning granularity and timelines, etc. None of this translate to the current solution – in particular, the lowest level actor (i.e., the implementation agent) cannot make design decisions here. Further, the waterfall model is generally considered an undesirable practice – hence, I am not sure why that would be a good inspiration. I believe the authors are making a different point about design before implementation, which needs to be stated clearly.

**Questions:**

1. Have you considered the possibility of going back to the design/planning phase if certain implementations become infeasible?

2. There is no discussion of limitations in the paper. It is unclear where this technique would and would not be suitable. For example, can we use this for code editing as well? Also, consider moving the unsolved problem analysis into the main paper.

3. Please comment on the selection of baselines and datasets in the evaluation.

---

> ### Author Response · Authors · 2024-11-16
>
> Thank you for your detailed comments. We will follow your advice and clarify all your points and questions.
>
> >Q: Why not pick  AgentCoder, MapCoder, CodeChain, WizardCoder, etc?
>
> A: We chose LDB, instead of AgentCoder, and so on, as LDB demonstrates state-of-the-art performance and is the most recent work. Including other methods would not provide a convincing comparison of the results. Furthermore, the term "code generation techniques" should ideally encompass both generation and debugging. We would like to clarify that some of the generation techniques mentioned by the reviewer also contain debugging steps.
>
> >Q:  Deeper analysis of the APPS and CodeContests dataset, as well as consider swebench.
>
> A:  Analysis of LPW and SLPW on the challenging APPS and CodeContests benchmarks is provided in **Appendix A.7.2**. This includes performance analysis, Pass@1 accuracy across different difficulty levels, Pass@1 accuracy vs. average token cost per program, as well as an exploration of possible failure reasons. We appreciate the suggestion about swebench, and we will incorporate it as a challenging benchmark.
>
>
>
> >Q:  Need reporting of variance or standard deviations.
>
> A: The variance for LPW and SLPW on HumanEval with GPT-3.5 is around **0.58 and 0.33**, based on three runs. The improvement observed in LPW and SLPW on HumanEval with GPT-3.5 is around **6.7%** compared to LDB. Additionally, LPW and SLPW show an average improvement of **4.7%** on the benchmarks in Table 1 with GPT-3.5, and **8.4%** on APPS and CodeContests with GPT-4o, compared to the current state-of-the-art LDB. LPW and SLPW outperform LDB across all benchmarks and LLMs, demonstrating consistently state-of-the-art performance.  Our result evaluation is consistent with previous works [1][2][3][4][5][6][7][8]
>
>
> >Q: Omit SD and SP on APPS and CodeContests?
>
> A: Since SD and SP already exhibit weaker performance on the simpler benchmarks, HumanEval and MPPP, compared to LDB, it is unlikely they would perform well on the more challenging benchmarks, APPS and CodeContests.
>
> >Q: The plots in Figure 4  should not have a non-zero value as the origin.
>
> A: In part (a) of Figure 4, With 0 iterations, the original accuracy is not zero, as it represents the accuracy before debugging. Plotting with a zero origin would compress all curves into the small top area, making it difficult to read. In part (b) of Figure 4, the percentage of solved problems in each difficulty category is crucial, which is clearly noted in the graph. The figure for part (b) with a zero origin is uploaded in the supplementary material.
>
> >Q: No reporting of cost-to-performance analysis.
>
> A: Figure 8 in the appendix presents a comparison of Pass@1 accuracy vs. the average token cost per program for LDB, LPW, and SLPW on the APPS benchmark with GPT4o. The cost-performance ratios are as follows: LDB achieves **0.32%** accuracy per thousand tokens, LPW achieves **0.35%**, and SLPW achieves **0.31%**. **LPW demonstrates the highest cost-effectiveness in terms of accuracy per token**
>
> >Q: There is insufficient discussion of related work.
>
> A: Thank you for your insightful suggestions regarding the recent works on the SWE-bench dataset and planning techniques. We will cite and incorporate them into our work.
>
> >Q: Comparison to human software-engineering processes.
>
> A: Thank you for your insightful comments on waterfall and scrum. LPW is not intended to replicate these methodologies; instead, it emphasizes a planning-first approach, separating plan generation from code implementation. We will revise our description as you suggested.
>
> >Q:  Going back to the design/planning phase if certain implementations become infeasible?
>
> A: Currently, we do not support reverting to the design/planning phase if certain implementations become infeasible, as we use multiple steps to ensure the accuracy of the generated plan, including plan verification and verification checks. The results in Table 10 indicate that the accuracy of the generated plan and its verification is around **93%** following our solution generation framework on the HumanEval with GPT-3.5.   Our primary concern is that this would lead to excessive token consumption. However, we recognize this as an interesting direction for future work.
>
> >Q: No limitation discussion. It is unclear where this technique would be suitable.
>
> A: Thank you for the suggestion to move the unsolved problem analysis to the main paper. We will incorporate this as part of the limitations analysis. The other limitation is how to efficiently translate the natural language solution into the code solution as plan and plan verification generation accuracy is typically higher than code generation accuracy.
> As we have mentioned in **Lines 97-98**, LPW and SLPW are specifically designed for text-to-code generation. However, code editing is also supported in LPW and SLPW, as they include code debugging capabilities.

---

> ### Author Response · Authors · 2024-11-25
>
> Thanks for the reviewer's reply
>
> We would appreciate it if you could elaborate further on your conclusion that "it is not ready for publication as is." Are there specific aspects of our response that remain unclear or do not address your concerns?
>
> If so, we would greatly appreciate your suggestions to help us improve both the revised manuscript and our future work.
>
> Thank you for your time and guidance.

---

> > ### Author Response · Authors · 2024-12-03
> >
> > We sincerely thank the reviewer for the thorough engagement.
> > ﻿
> > We have updated our manuscript based on your suggestions, and some experiments are still ongoing as the deadline constrain. We kindly ask whether your main concerns have been addressed and if there are any other questions of interest.
> > ﻿
> > We deeply appreciate your time and thoughtful involvement.

---

### Official Review · Reviewer_N5sF · 2024-10-26

**Soundness:** 1
**Presentation:** 3
**Contribution:** 3
**Rating:** 3
**Confidence:** 4

**Summary:**

This paper proposes an LLM "programming workflow" that the authors dub LPW.
The key idea is to decompose code generation into a sequence of distinct tasks:
- "Planning" how to solve the problem (e.g. coming up with the "architecture" of the solution)
- Constructing the initial code
- If the initial code does not pass the tests:
  - Compare the execution trace to the "plan" to find the bug (note this treats the plan as gold truth)
  - Fix the program
In addition, given the importance of the "plan", the authors include a verification step where an LLM steps through the plan and ensures an implementation thereof will match the specification.

In terms of results, the experiments focus on standard, self-contained Python programming benchmarks such as APPS, HumanEval, MBPP and CodeContests. The authors claim to improve the pass rate compared to several benchmarks including simple self-debugging and one prior agentic approach, but I do not have much confidence in the veracity of their claims (see weaknesses).

**Strengths:**

This is an important research area, and projects such as this may have a big impact for practitioners.
I believe this paper also includes some novel and interesting ideas, such as comparing the plan to the execution trace during debugging (to help the LLM reason about what went wrong), as well as (in the "sampling" version, SLPW) using a technique from the bandit literature (UCB) to select the set of plans to attempt to implement based on the number of unit tests that the plan is consistent with.
The methodology is mostly clear (see weaknesses) and the writing is good; the authors also do a good job calling out to related work

**Weaknesses:**

Unfortunately, I was gravely disappointed by the analysis of the experiments in this paper.
In their current form, I do not believe they are meaningful in any real sense - i.e., I don't think they accurately assess how this approach compares to simple baselines like just sampling IID from the model with a higher temperature. Reading lines 338-345 I believe anyone familiar with the code generation literature would immediately realize that the comparisons to the other methods will be unfair.

In particular, the approach fundamentally involves drawing multiple samples from the model - one to construct the plan, one to verify it, one to generate the code, one to compare the trace to the plan (if the code failed), and one to fix the program - and yet the authors label the pass rate of the *final* program as "pass@1", which is incredibly misleading as pass@1 has an established meaning in the literature - the expected pass rate of *one* sample from the model.
Some of these additional calls to the LLM can be coalesced, but some of them cannot.
In particular, anything before execution can obviously not be coalesced with anything that comes after.
As a first order of approximation, it thus seems more fair to compare "pass@1" from LPW with pass@2 from the baseline, but even this does not adequately capture the fact that you have to sample a bunch of additional tokens to construct and verify the plan (in addition to the program).
This is made even worse in the "sampling" variation of the method, where at each point you branch (and, thus, end up with a whole tree of samples from the model).

Evaluating these agentic systems fairly is hard, but there has been much prior work in the literature that has at least **attempted** to account for the additional costs involved (e.g. [1, 2]); sadly this paper is not one of them.
There is only one instance in which I can imagine that the current evaluation strategy would make sense, and that is if LPW was able to solve programs that the baseline *never* could, in which case cost would be irrelevant.
There may or may not be a handful of such tasks in CodeContests, but there certainly are none in APPS, HumanEval or MBPP considering the extensive contamination these aging datasets have been exposed to.

I want to emphasize that there are some interesting nuggets in this paper, and I commend the authors for their effort.
Nonetheless, I truly feel that the experiments are so unfair that they are meaningless, and thus have no choice but to reject the paper in its current state.
I encourage the authors to think more carefully about how to fairly evaluate their system against the different baselines in a way that takes the different costs into account; note that doing so would not require running any more experiments, just changing how you analyze the results (assuming you didn't throw the data away, that is). Concretely, one way to approach this might be to compare pass rates versus the total number of tokens sampled from the model, as proposed in [2].

[1] Chen, Xinyun, et al. "Teaching large language models to self-debug." arXiv preprint arXiv:2304.05128 (2023).
[2] Olausson, Theo X., et al. "Demystifying gpt self-repair for code generation." arXiv preprint arXiv:2306.09896 (2023).

**Questions:**

- Have you considered some way of quantifying the computational cost (e.g. the number of samples drawn from the method) of each method?
- If you took the cost of each method into account, how would LPW and SLPW compare against the baselines?
- Given the relative age of the benchmarks used, what impact (if any) do you think data leakage/contamination may have on your conclusions?

---

> ### Author Response · Authors · 2024-11-20
>
> Thank you for your response and constructive suggestions. However, there are still some misunderstandings from the reviewer. We hope our reply will clarify these misunderstandings and help enhance our work.
>
> Thank you for acknowledging Pass@1 as a validated experimental evaluation metric. We are glad to know that we have addressed one of the main concerns.
>
> >Q: The baseline challenging the conclusion of this paper,
>
> A: Regarding the baseline challenging the conclusion of our paper, we disagree with the reviewer's idea. The approaches of generating multiple sample programs, selecting one based on visible tests or other metrics, and then evaluating it on hidden tests have been thoroughly investigated in prior work [1][2][3][4], as explicitly noted on Lines 45–48. Comparison between methods relying on sampling and visible test-based selection [1] versus those incorporating debugging (e.g., SD and LDB), SD and LDB reveal substantial performance improvements. Including non-SOTA methods in the comparison would not yield a compelling or meaningful evaluation.
>
> Besides, the Pass@1 is not our own definition. It is the standard metric that we both agree on in the discussion above.
>
> >Q: if LPW on average takes X samples to solve a task, and SD takes Y < X, why don't you first sample X- Y programs, validate them against the visible tests, and then pick one uniformly at random to self-debug if needed
>
> A: The reviewer appears to misunderstand LPW and subsequently draws an incorrect conclusion. We hope this clarification will address the misunderstanding and underscore the significance of LPW.
>
> LPW is consistent with LDB and SD where only **one code** is generated, and **this code** is iteratively debugged throughout the entire pipeline. Therefore, the reviewer’s concern about X programs in LPW and Y programs in SD is not applicable, as this does not reflect the operation of LPW and SD as described in our paper. In both cases, **a single program** is generated initially and iteratively debugged. Unlike SD and LDB, LPW introduces a novel plan verification, which is treated as the intended natural language solution to guide initial code generation (producing only **one code**) and subsequent debugging (still operating on the **single initially generated program**). We present various results, including Pass@1 accuracy, performance after adding more visible tests, some ablation studies, an evaluation of plan verification accuracy, and its contribution to correct code generation, in both the main text and the appendix to emphasize its significance and its effects in the code generation process.
>
> In contrast, SLPW involves multiple plans, plan verifications, and codes (Please note in SLPW, only one code that passes visible tests is selected to validate on hidden tests). As we clearly noted, SLPW is a sampling variant of LPW, requiring additional tokens to achieve the highest accuracy. However, this is a separate contribution and does not detract from LPW’s core innovation, where plan verification plays a critical role in both initial code generation and subsequent debugging. Building on LPW's strong performance, we proposed SLPW, which achieves state-of-the-art results while highlighting the utility of LPW’s plan verification mechanism.
>
> >Q: To illustrate my concerns in plain language: Suppose I have a model that generates an entirely correct program (passing all tests) 50% of the time, and an entirely incorrect program (passing no tests) 50% of the time. If you truly take only one sample from the model, then you would report a pass@1 of 50%. If you took two samples, validated them against the visible tests and picked one (if any) that passed those to be validated against the hidden tests, then you would get a pass@1 (under the definition used by the authors) of 75%. Thus, simply testing against the visible tests can do a lot of the heavy lifting here, and since they do not attempt to balance the budgets the current experiments do not separate out this noise from whatever signal there is.
>
> A:   Thank you for your example to clarify your concerns. Again, when LPW is compared with LDB and SD, the situation is different, as only a single code is involved. For SLPW, only one code that passes the visible tests is chosen to validate the hidden test, and the probability of the selected code passing the hidden test remains 50%. While multiple generated codes can increase the likelihood of passing the visible tests, as illustrated in the reviewer's example, they do not affect the probability of passing the hidden test for the single selected code.

---

> ### Author Response · Authors · 2024-11-20
>
> >Q: investigate the cost (in tokens) vs. model performance
>
> A: Thank you for the reviewers' valuable insights regarding performance vs. token usage. As the reviewer mentioned, LDB exhibits relatively flat performance curves, whereas the LPW and SLPW demonstrate steeper curves. As highlighted earlier in our discussion of the APPS benchmark, LPW and SLPW require more token resources to achieve 50% accuracy but subsequently use fewer tokens to attain higher accuracy levels. Again, we agree with the reviewer that comparing the pass@k accuracy may give an incomplete picture of model performance when comparing models with different token consumption. One way to compare is the cost-performance ratios, as shown in Figure 8, as well as incorporating the reviewer’s suggestion of a graph illustrating pass rate variations relative to token consumption
>
> We will include a detailed graph illustration of the pass rate variations with token consumption for each method in the appendix. Additionally, we will explain these findings in both the main text and the appendix. We greatly appreciate your thoughtful feedback—thank you!
>
> >Q: Finally, I would encourage the authors to consider following good statistical practice by showing variance or confidence intervals whenever they are displaying averages, as is done in this figure. In particular, in this figure both the x and the y are averages, and so each point should be surrounded by confidence intervals along both axes.
>
> A: We appreciate your insightful suggestions and will follow it to make good statistical practice.
>
> >Q: Data leakage:
>
> We appreciate the reviewer for clearly identifying the issue of data leakage and providing valuable suggestions for future work. To address this, we will incorporate the suggested benchmarks to present more convincing results and eliminate concerns about data leakage.
>
> [1] Bei Chen, Fengji Zhang, Anh Nguyen, Daoguang Zan, Zeqi Lin, Jian-Guang Lou, and Weizhu Chen. Codet: Code generation with generated tests. In Proceedings of the 11th International Conference on Learning Representations, ICLR, pp. 1–19, 2023a
>
> [2]Tianyi Zhang, Tao Yu, Tatsunori Hashimoto, Mike Lewis, Wen-tau Yih, Daniel Fried, and Sida
> Wang. Coder reviewer reranking for code generation. In Proceedings of the 40th International
> Conference on Machine Learning, ICML, pp. 41832–41846, 2023.
>
> [3]Ansong Ni, Srini Iyer, Dragomir Radev, Veselin Stoyanov, Wen-tau Yih, Sida Wang, and Xi Victoria
> Lin. Lever: Learning to verify language-to-code generation with execution. In Proceedings of the
> 40th International Conference on Machine Learning, ICML, pp. 26106–26128, 2023.
>
> [4]Mark Chen, Jerry Tworek, Heewoo Jun, Qiming Yuan, Henrique Ponde de Oliveira Pinto, Jared
> Kaplan, Harri Edwards, Yuri Burda, Nicholas Joseph, Greg Brockman, et al. Evaluating large
> language models trained on code. arXiv preprint arXiv:2107.03374, 2021.

---

> > ### Comment · Reviewer_N5sF · 2024-11-22
> >
> > I thank the authors for their ongoing willingness to address my concerns.
> >
> > I am glad that we agree that visible tests play an important role in this line of work. However, it is apparent that we have still not reached an agreement on the impact of this on the validity of your results; in particular, you say that:
> >
> > > While multiple generated codes can increase the likelihood of passing the visible tests, as illustrated in the reviewer's example, they do not affect the probability of passing the hidden test for the single selected code.
> >
> > This is patently false, because passing the visible tests and passing the hidden tests are obviously not uncorrelated events. In fact, evidence for this is given in Table 8 of the submission: out of the programs that pass the visible tests, most of them also pass the hidden tests. So, drawing multiple samples (to match the sample complexity of LPW, counting not just the initial program but the plan generation etc), testing them against the visible tests and then picking the one that passes the most of them is a much stronger baseline than the one you have considered in the paper.
> >
> > Of course the extent to which passing the visible and the hidden tests are correlated events depends on the coverage of the visible tests, which is exactly why it is important to have a simple baseline that measures only the effect of *this* on the pass rates. Unlike what the authors seem to think, the point of baselines is *not* to have more "non-SOTA" results cluttering the results, but to separate the effect of how you model the task vs. the effect of your proposed methodology. (And, for the record, if you come up with a simple baseline that shows that the conclusions of some prior work were erroneous, that would be a much more valuable contribution to the community than increasing some "SOTA" by a few %; so it is always worthwhile to spend time optimizing your baselines, regardless of what prior work has done.)
> >
> > Furthermore, I want to emphasize that I do *not* misunderstand neither LPW nor SD when I speak of drawing multiple samples; I am well aware that you are only drawing 1 sample at a time. The point is that combining, for example, SD with multiple samples is an *obvious* extension that would be necessary to separate the effect of--for example--generating the plan vs. the effect of the additional compute you require to solve the task. Without doing this, I as the reader have no way of telling whether your (substantially more costly) method is worth the additional compute or not.
> >
> > The changes promised by the authors are good, and will improve the paper substantially. I encourage the authors to make those changes easily visible in the paper (e.g. by coloring them in blue), and will certainly reconsider my score once the new draft has been made available. However, unless progress is made on making the evaluation more fair to the baselines and other methods, I do not expect to raise my score substantially.

---

> ### Author Response · Authors · 2024-12-02
>
> We have revised our manuscript based on your valuable feedback.
>
> Below are some recent experimental results that could not be included in the manuscript due to deadline constraints. We will add them to the final version.
>
> 1): The performance of various approaches on the LiveCode benchmark, assessed using a subset of 140 sub-problems with GPT-4o.
>
> || LiveCode|
> |:-|:-:|
> |Baseline|45.7±0.6 |
> |LDB|54.3 ±0.3|
> |LPW|59.3 ±0.6 |
>
> LPW outperforms LDB by 5% in Pass@1 accuracy and surpasses the Baseline by approximately 15% in Pass@1 accuracy.
>
> 2): We also introduce Repeated Sampling as a comparative method. For each problem, it repeatedly samples program solutions from the LLM until either the token consumption surpasses that of SLPW, or a solution passes the visible test and is then validated on hidden tests. The preliminary results with GPT-4o are as follows:
>
>
> ||APPS|CodeContests|
> |:-|:-:|:-:|
> |Repeated Sampling|51.1±0.9|32.0±0.5|
> |LDB|53.2±0.3|29.3±0.3|
> |LPW| 62.6±0.3|34.7±0.3|
> |SLPW|64.0±0.3|35.3±0.3|
>
>
> Repeated Sampling is allocated the same token budget as SLPW, while its accuracy remains lower than LPW and SLPW, highlighting the benefits of plan and plan verification in generating high-quality initial code and subsequent refinements.
>
> We also encourage the reviewers to refer to Figures 10 and 11 in Appendix A.8. We hope these will help address the concerns regarding cost performance.
>
> We sincerely thank the reviewers once again for the valuable feedback and hope that our responses adequately address the reviewer's concerns.

---

### Official Review · Reviewer_DqDY · 2024-10-30

**Soundness:** 3
**Presentation:** 3
**Contribution:** 2
**Rating:** 5
**Confidence:** 3

**Summary:**

This paper propose an LLM programming workflow (LPW). The key point of this framework is the plan propose phase and the plan refinement based on visible tests. Then LPW will implement the code based on the verified plans. The refined plan also can help refine the code when the generated code cannot pass the visible tests. It also presents SLPW, which is  a sampling variant to generation multiple solutions and refine as necessary. The results show improvements on benchmarks like HumanEval, MBPP, APPS and CodeContest.

**Strengths:**

1. This paper presents a well-designed framework. The technical details are solid and easy to reproduce.
2. This paper is presented clearly, illustrating the methods greatly.
3. The authors conduct a series of experiments, which is of great soundness.

**Weaknesses:**

1. The key point of LPW is the plan proposal and refinement based on visible tests. However, I'm not fully convinced about the refinement capability of plans since it is fully dependent on LLMs. The authors ask the LLM to dry run the plan through the test input, which in essence is a reasoning problem. It would require the LLM to be great at reasoning that it can dry run the operations and get the 'real output' and compare it with the 'grounded output'. Otherwise, it could derive wrong results even when the plan is correct (hallucination).
If the dry run on plans gives incorrect results, it will fail in result comparison and lead to refinement on correct plans and possibly ends up with a wrong plan. Therefore, I'm a bit curious about the accuracy of the LLM plan dry run, and the accuracy of LLM verficiation checks. Also, what is the authors' insight to improve this accuracy to alleviate the hallucination problem? More ablation study on this would be helpful. Also since the plan acts an important role in LPW, it would be beneficial to sho how many iteration number it takes to refine the plan.

2. The other part of this paper has limited novelty: compared to the related works mentioned in this paper, it seems the paper aggregates these methods to achieve high performance. While this is valuable in real-world applications, the lesson from this is limited. More illustrations on the design choice would be valuable.

3. For SLPW, it is similar to 'pass@k' version of LPW. The author use the UCB algorithm to help decide whether to explore. I'm curious about the effectiveness of the UCB methods here compared to simply selecting the code with the most passed tests.

**Questions:**

1. Could you illustrate the accuracy of the LLM plan dry run and verification checks?
2. Compared to verify the code intermediate results, what is the benefit of verifying plans directly? How to guarantee the intermediate results generated by LLM reasoning is correct?
3. Could you illustrate more on why choosing the UCB algorithms over simply selecting the code passing the most visible tests?

---

> ### Author Response · Authors · 2024-12-03
>
> We appreciate the reviewer's questions and constructive feedback
> ﻿
> As the deadline approaches, we kindly ask whether our responses have resolved your concerns.
> ﻿
> We sincerely appreciate your time and thoughtful involvement.

---

### Official Review · Reviewer_cTjF · 2024-11-04

**Soundness:** 3
**Presentation:** 3
**Contribution:** 3
**Rating:** 8
**Confidence:** 5

**Summary:**

The paper introduces the LLM Programming Workflow (LPW) and LPW with Sampling (SLPW), which is a 3-stage agentic framework to solve function-level NL-to-Code problems like HumanEval and MBPP. The 3 stages are:

1. **Plan Generation**: the backbone LLM is prompted to generate a step-by-step natural language plan for implementation
2. **Plan Verification & Refinement**: the backbone LLM is prompted to simulate the steps in the natural language plan on visible tests, compare the simulated output against the expected output, and (optionally) self-refine the plan until a self-verified plan is generated.
3. **Iterative implementation & Refinement**: the backbone LLM is prompted to implement the solution given the self-verified plan, and (optionally) self-refine the implementation given various feedback: failed hidden test case, expected output, execution trace, explanation of failure, and revised-and-verified plan.

Experiments demonstrate that LPW and SLPW outperform Self-Planning [2], Self-Debugging [3], and LDB [4] when using GPT-4o, GPT-3.5, Llama-3, and Phi-3 as the backbone model.


[1] [Parsel: Algorithmic Reasoning with Language Models by Composing Decompositions](https://arxiv.org/abs/2212.10561)

[2] [Self-planning Code Generation with Large Language Models](https://arxiv.org/abs/2303.06689)

[3] [Teaching Large Language Models to Self-Debug](https://arxiv.org/abs/2304.05128)

[4] [Debug like a Human: A Large Language Model Debugger via Verifying Runtime Execution Step-by-step](https://arxiv.org/abs/2402.16906)

[5] [NExT: Teaching Large Language Models to Reason about Code Execution](https://arxiv.org/abs/2404.14662)

[6] [CRUXEval: A Benchmark for Code Reasoning, Understanding and Execution](https://arxiv.org/abs/2401.03065)

**Strengths:**

## Novelty

The paper successfully integrates previous working methods, Self-Planning [2] and Self-Debugging [3], into the LPW framework. In addition, LPW novelly introduces the "Plan Verification & Refinement" stage to iteratively refine the quality of the plan.

## Effectiveness

The LPW framework demonstrates substantial improvements over previous agentic frameworks like Self-Planning [2], Self-Debugging [3], and LDB [4].



[1] [Parsel: Algorithmic Reasoning with Language Models by Composing Decompositions](https://arxiv.org/abs/2212.10561)

[2] [Self-planning Code Generation with Large Language Models](https://arxiv.org/abs/2303.06689)

[3] [Teaching Large Language Models to Self-Debug](https://arxiv.org/abs/2304.05128)

[4] [Debug like a Human: A Large Language Model Debugger via Verifying Runtime Execution Step-by-step](https://arxiv.org/abs/2402.16906)

[5] [NExT: Teaching Large Language Models to Reason about Code Execution](https://arxiv.org/abs/2404.14662)

[6] [CRUXEval: A Benchmark for Code Reasoning, Understanding and Execution](https://arxiv.org/abs/2401.03065)

**Weaknesses:**

## Novelty

Most design choices in LPW have been established and analyzed in previous works: the 1st stage "Plan Generation" is drawn from Self-Planning [2], and the 3rd stage "Iterative implementation & Refinement" follows various prior works [3, 4, 5], especially regarding the feedback mechanisms.

The main novel contribution of LPW lies in its 2nd phase, "Plan Verification & Refinement," making the paper’s overall novelty contingent on the soundness and effectiveness of this one component.



## Soundness

The self-verification process is essentially the LLM simulating the natural language plan on a visible input test case and trying to predict the output. While conceptually appealing, this task is inherently challenging for LLMs as shown by the CRUXEval benchmark [6].

In CRUXEval, the models are tasked to predict the output given an input and a very simple program (no imports, limit on input length, limit on program length). In the setting of LPW, the models are dealing with a more challenging case of predicting the outputs given an arbitrarily complex input and a natural language plan.

For example, if we replace the visible input test case in Figure 5, [-3, -4, 5] with a lengthy list of large integers like [2890, 4719, 1230, 2309, 240, 2190, 34789, 91, 12, 928, ...], it's hard to imagine that the LLM will be able to predict the correct output, even though the nature language plan is simple enough.

It'd be highly beneficial for the author to conduct an in-depth analysis of the quality and reliability of the self-verification process. Some key metrics can be:

1. **Type I Error Rate**: What's the percentage of plans that are correct but get verified as incorrect?
2. **Type II Error Rate**: What's the percentage of plans that are incorrect but get verified as correct?

The current ablation study, represented by LPW-C, ablates both the 1st and 2nd stages, which does not isolate the specific effectiveness of the "Plan Verification & Refinement" stage.

Additionally, separating the effects of "Plan Refinement" and "Plan Verification" could provide further valuable insight. For instance, what would the performance be if the model only performed self-critique and self-refined the plan without simulating it on visible test cases to predict outputs?

Given these observations, the current version of the paper does not fully justify the soundness of its main novel contribution, "Plan Verification & Refinement."

This aspect has the most impact on my Overall Rating, Soundness Rating, and Contribution Rating. I would be glad to reconsider and potentially raise my scores if the authors could address the above concerns.



## Significance

Although LPW demonstrates superior accuracy compared to existing agentic frameworks, it also requires more model calls, potentially increasing the time and financial costs associated with solving each problem. This may impact LPW’s practical applicability in real-world settings where efficiency is critical.

It would be beneficial if the authors could provide an analysis comparing the efficiency and cost of LPW and SLPW with baselines such as Self-Debugging, LDB, LPW-S, and LPW-C. In terms of cost-performance ratio, which framework is the best?

This aspect has a secondary impact on my Overall Rating and Contribution Rating. I would be glad to reconsider and potentially raise my scores if the authors could address the above concerns.



[1] [Parsel: Algorithmic Reasoning with Language Models by Composing Decompositions](https://arxiv.org/abs/2212.10561)

[2] [Self-planning Code Generation with Large Language Models](https://arxiv.org/abs/2303.06689)

[3] [Teaching Large Language Models to Self-Debug](https://arxiv.org/abs/2304.05128)

[4] [Debug like a Human: A Large Language Model Debugger via Verifying Runtime Execution Step-by-step](https://arxiv.org/abs/2402.16906)

[5] [NExT: Teaching Large Language Models to Reason about Code Execution](https://arxiv.org/abs/2404.14662)

[6] [CRUXEval: A Benchmark for Code Reasoning, Understanding and Execution](https://arxiv.org/abs/2401.03065)

**Questions:**

## Clarification Questions

In Section 7, the paper states that previous iterative refinement frameworks **"encounter difficulties when the initial code substantially deviates from the original intent"**.

What are the commonalities among problems that LPW solved but LDB failed? Can the LDB failures be attributed to the quality of the initial plan?

It would be better if the authors could provide an error distribution analysis of those problems and demonstrate some examples.

---

> ### Author Response · Authors · 2024-11-19
>
> Thank you for your thoughtful feedback and encouragement! We greatly appreciate your insights and will present a comprehensive study that thoroughly examines both the benefits and limitations of self-verification and your comments are invaluable in guiding us toward that goal!
>
> >Q: Please consider adding the detailed experiment setting, especially the human annotation process, and analysis of the results to the main content as an ablation study.
>
> Thanks for your suggestion, we will give all the human annotation details in the appendix with a new section and also add the result to the ablation study.
>
>
> >Q: In addition, as mentioned in my original comment, it is still a concern that the complexity of test cases can affect the accuracy of the plan verification process. This seems to be the biggest limitation of the proposed self-verification technique. It would be ideal for the author to explore and inform the community of such a limitation.
>
> A: We sincerely appreciate your thoughtful feedback and hope that the experiments included in the Appendix will help to address your concerns. Appendix Table 10 presents the percentage of problems where the LLM successfully generates the valid plans and plan verifications in the solution generation phase - **94.5%** for LPW and **95.1%** for SLPW - and percentage of problems where LLM-generated plans are manually classified as correct — **92.7%** for LPW and **93.9%** for SLPW (​​considering no plan cases) — and the percentage where LLM-generated plan verifications are manually classified as correct —**92.7%** for LPW and **93.3%** for SLPW (considering no plan verification cases) — in the HumanEval benchmark with GPT-3.5. **In general the plan verification performs well within the solution generation framework**.
> From Table 10, **7.3%** problems for LPW and **6.7%** for SLPW cannot be solved due to the absence of validated plans and plan verifications. **One reason, as highlighted by the reviewer, is that complex test cases present challenges for LLMs during plan verification, leading to no validated plan verification**. We will provide a more detailed analysis in the Appendix, along with illustrative examples to better show these cases. Thanks for your insight and suggestions.
>
> We note that for the HumanEval benchmark, **92.7%** of generated plan verifications for LPW and **93.3%** for SLPW are correct. These percentages exceed the respective code accuracy rates for LPW and SLPW. As shown in Appendix Table 11, over **95%** correct plan verifications directly contribute to correct code generation, while incorrect plan verifications almost always result in incorrect code. These results demonstrate the reliability of the solution plan and plan verification generated within the solution generation framework.
>
> >Q: Please consider adding the LPW-V and SLPW-V results to the main content i.e. Table 4.
>
>  A： Thank you for your suggestion about LPW-S and LPW-V; we will incorporate it into Table 4 as recommended.
>
> >Q: Figure 8 is insightful. Please consider having a dedicated section in the Appendix.
>
> Thank you for your suggestion; we will dedicate an entire section to discussing the cost, as recommended, including analyses of HumanEval, MBPP, vanilla Self-Debugging, and Self-Planning. We choose APPS as an example due to its challenging nature for LLMs.
>
>
> >Q: This analysis makes sense. Sorry for the confusion, I think the actual question should be how LPW solves problems that LPW-V can't solve. Such case analysis will demonstrate how the self-verification technique improves a farfetched initial plan into a more reasonable one, making the code generation process, and thus the debugging process, easier.
> Please consider having a dedicated section in the Appendix for this analysis.
>
> A: Thank you for clarifying your question, and we apologize for the misunderstanding. An example can be found in the 120th problem of HumanEval, illustrated in Figure 5. LPW successfully solves this problem as plan verification confirms that the return array must be in ascending order, based on the visible tests. This plan verification effectively guides the debugging step to refine the code correctly. Without plan verification (LPW-V), the unverified plan merely guides the code to return the first K elements, leading to failure. As suggested, we will dedicate an entire section to discussing how self-verification influences the whole pipeline.

---

> > ### Comment · Reviewer_cTjF · 2024-11-23
> >
> > Thank the authors for the response. Please let me know after you update the paper, I will raise my scores accordingly.

---

> ### Author Response · Authors · 2024-12-02
>
> We have updated our manuscript based on your suggestions and sincerely thank you once again for your constructive feedback.

---

> > ### Comment · Reviewer_cTjF · 2024-12-02
> >
> > Thanks, I have raised my soundness, contribution, and overall ratings accordingly.

---

### Author Response · Authors · 2024-12-02
**Manuscript Updating**

We sincerely appreciate the constructive feedback provided by the reviewers, and we have incorporated their suggestions into the revised manuscript. All updates are highlighted in blue.

The modifications include:

1. Change the discussion on the inspiration behind LPW (Lines 97–99).
2. Update the results in Tables 1 and 2 to include standard deviations.
3. Move LPW-V into the ablation study in the main text, adding the corresponding description (Lines 428–474).
4. Summarize the cost-performance analysis on Lines 384–386 with a reference to the detailed discussion in Appendix A.8.
5. Provide a detailed cost-performance analysis in Appendix A.8.
6. Include a discussion on limitations in Appendix A.9.
7. Supplement the relationship between plan and plan verification on Lines 996–999 and 1006–1008.
8. Relocate the performance analysis across different difficulty levels from the main text to Appendix A.4 to save space (Lines 916-933).

---

### Note · Authors · 2024-12-13

I have read and agree with the venue's withdrawal policy on behalf of myself and my co-authors.